# StateFlow: Enhancing LLM Task-Solving through State-Driven Workflows

**Yiran Wu**
Pennsylvania State University
yiran.wu@psu.deu

**Tianwei Yue**
MathGPTPro
tianwei.yue@mathgptpro.com

**Shaokun Zhang**
Pennsylvania State University
shaokun.zhang@psu.edu

**Chi Wang**
Microsoft Research Redmond
wang.chi@microsoft.com

**Qingyun Wu**
Pennsylvania State University
qingyun.wu@psu.deu

## Abstract

It is a notable trend to use Large Language Models (LLMs) to tackle complex tasks, e.g., tasks that require a sequence of actions and dynamic interaction with tools and external environments. In this paper, we propose StateFlow, a novel LLM-based task-solving paradigm that conceptualizes complex task-solving processes as state machines. In StateFlow, we distinguish between "process grounding" (via state and state transitions) and "sub-task solving" (through actions within a state), enhancing control and interpretability of the task-solving procedure. A state represents the status of a running process. The transitions between states are controlled by heuristic rules or decisions made by the LLM, allowing for a dynamic and adaptive progression. Upon entering a state, a series of actions is executed, involving not only calling LLMs guided by different prompts, but also the utilization of external tools as needed. Our results show that StateFlow significantly enhances LLMs' efficiency. For instance, StateFlow achieves 13% and 28% higher success rates compared to ReAct in InterCode SQL and ALFWorld benchmark, with 5× and 3× less cost respectively. We also show that StateFlow can be combined with iterative refining methods like Reflexion to further improve performance.

## 1 Introduction

LLMs have increasingly been employed to solve complex, multi-step tasks. Specifically, they have been applied to tasks that require interactions with environments (Yang et al., 2023a; Yao et al., 2022a; Shridhar et al., 2020; Zelikman et al., 2022) and those tasks that can benefit from utilizing tools such as web search and code execution (Mialon et al., 2023; Wu et al., 2023b; Wang et al., 2023c). In approaching these tasks, there is typically a desired workflow, or plan of actions based on heuristics that could improve the efficiency of task solving (Kim et al., 2023; Wu et al., 2023a). A common practice in the context of LLMs, such as ReAct (Yao et al., 2022b) and the vast customization of GPTs, is to write a single prompt that instructs the models to follow a desired procedure to solve the task (Dohan et al., 2022). The LLM is called iteratively with the same instruction, along with previous actions and feedback from tools/environments. This relies on LLMs' innate capability to determine the current task-solving status and perform subsequent actions autonomously. Despite the impressive abilities of LLMs, it is still unrealistic to expect LLMs to always judge the status of current progress correctly. It is also almost impossible to reliably track these judgments and their decisions of subsequent action trajectory. Given these considerations, we pose the research question: How can we exert more precise control and guidance over LLMs?

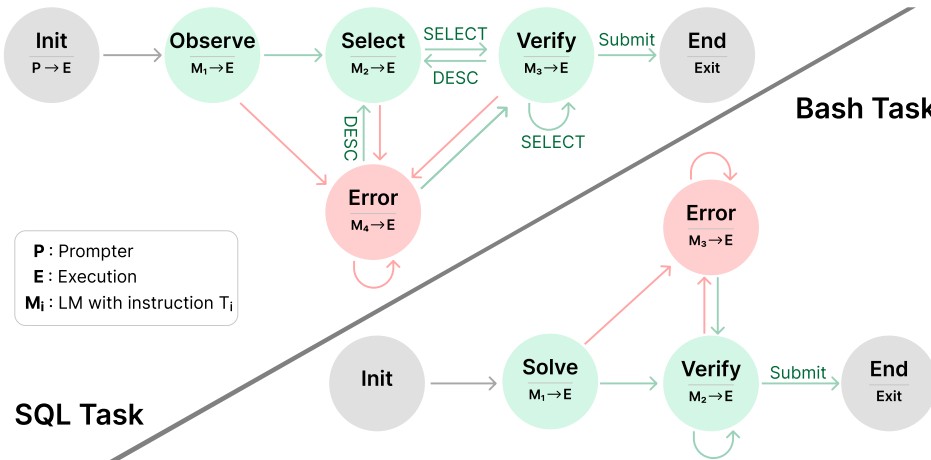

Figure 1: The `StateFlow` models for the SQL and Bash task. `Init` and `End` state are basic components of state machines, and states like `Observe`, `Solve`, `Verify`, `Error` can be adaptable across various tasks. When reaching a state, a sequence of output functions defined is executed (e.g., $M_i \rightarrow E$ means to first call the model and then call the SQL/Bash execution). Execution outcomes are indicated by red arrows for failures and green for successes. Transition to different states is based on specific rules. For example, at a success 'Submit' command, the model transits to `End` state.

In this paper, we propose `StateFlow`, a new framework that models LLM workflows as state machines. Finite State Machines (FSMs) (Mealy, 1955; Moore et al., 1956) are used as control systems to monitor practical applications, such as traffic light control (Wagner et al., 2006). A defined state machine is a model of behavior that decides what to do based on current status. A state represents one situation that the FSM might be in. Drawing from this concept, we want to use FSMs to model the task-solving process of LLMs. When using LLMs to solve a task with multiple steps, each step of the task-solving process can be mapped to a state. For example, to solve the InterCode (Yang et al., 2023a) SQL task, a desired procedure is to first gather information about the tables and columns in the database, then construct a query to retrieve required information, and finally verify the task is solved and end the process. We can convert this workflow to a set of states (See upper left of Figure 1). Within each state, we define a sequence of output functions, which will be called upon entering the state. The output functions would take in the context history and output a new context to be appended to the history, which can be a tool, an LLM with a specific instruction, or a prompter. Based on the current state and context history, the `StateFlow` model would determine the next state to transit to. The task-solving process progresses by transitioning through different states and calls to corresponding output functions, and ends until a final state is reached. Thus, `StateFlow` enhances control over the task-solving process and seamlessly integrates external tools and environments.

We evaluate `StateFlow` on the SQL task and Bash task from the InterCode (Yang et al., 2023a) benchmark and the ALFWorld Benchmark (Shridhar et al., 2020). The results demonstrate the advantages of `StateFlow` over existing methods in terms of both effectiveness and efficiency. With GPT-3.5, `StateFlow` outperforms ReAct by 13% on InterCode SQL task and 28% on ALFWorld, with 5× and 3× less LLM inference cost respectively. `StateFlow` is orthogonal to methods that iteratively improve future attempts using feedback based on previous trials (Shinn et al., 2023; Madaan et al., 2024; Prasad et al., 2023). Notably, we show that `StateFlow` can be combined with Reflexion (Shinn et al., 2023), improving the success rate on ALFWorld from 84.3% to 94.8% after 6 iterations.

Our main contributions are the following: (1) We introduce `StateFlow`, a paradigm that models LLM workflows as state machines, allowing better control and efficiency in LLM-driven task solving. We provide guidelines on how to build with the `StateFlow` framework and illustrate the building process through a case study. (2) We use three different tasks to

illustrate the effectiveness and efficiency of `StateFlow`, with improvement in performance and a 3-5× cost reduction. We also perform an ablation study to provide deeper insights into how different states contribute to the performance of `StateFlow`. (3) We show that `StateFlow` can be combined with iterative refining methods to further improve performance.

## 2 Background

**Finite-state Machines**   We first introduce state machines, which we will use to formulate our framework. A finite state machine (automaton) is a mathematical model of a machine that accepts a set of words or string over an input alphabet $\Sigma$ (Hopcroft et al., 2001; Carroll & Long, 1989), where read of symbols would lead to state transitions. The automaton would determine whether the input is accepted or rejected. A basic Deterministic Finite-state Machine (DFSM) is a Deterministic Finite-state **acceptor**, usually used as a language recognizer that determines whether an input ends in an accept state. An acceptor is not suitable in our modeling of LLM generations, where we want to determine actions to be performed and produce outputs in between states. Instead, we base our model on a **transducer** finite-state machine, which is a sextuple $\langle \Sigma, \Gamma, S, s_0, \delta, \omega \rangle$ (Rich et al., 2008), where $\Sigma$ abd $\Gamma$ are the input and output alphabet (finite non-empty set of symbols), $S$ is a finite non-empty set of states, $\omega$ is the output function, $s_0$ is the initial state, $\delta$ is the state transition function ($\delta : S \times \Sigma \to S$), and $F$ is the set of final states.

## 3 Methodology

In this section, we first define the `StateFlow` model. We then provide a general guideline for constructing `StateFlow` model and illustrate with a detailed case study.

### 3.1 StateFlow

In a finite state machine, a state carries information about the machine's history, tracking how the state machine has reached the present situation (Wagner et al., 2006). It is feasible to conceptualize the task-solving process with LLMs as a state machine. Different from traditional FSMs, `StateFlow` doesn't have the concept of input tape but solely depends on the context history, which is a cumulative record of all past interactions. `StateFlow` employs a set of instructions $T = T_1, T_2, ..T_i$ to guide the language model generation at different states. This is equivalent to constructing a set of LLM agents $p_\theta^{T_i}$ with a specific instruction $T_i$. This dynamic prompting approach ensures that the language model receives the most relevant guidance at each state, improving its ability to focus on a specific step. Following the definition of finite state machines, we formulate a `StateFlow` model to be a sextuple $\langle S, s_0, F, \delta, \Gamma, \Omega \rangle$ and explain each of the components under the LLM scenario:

**States** $S$   A state encapsulates the current status of a running process, essentially an abstraction of the context history. Upon entering a state, a predefined set of actions is executed. For example, entering an `error` state implies the process encounters an issue, triggering the execution of predetermined error-handling actions.

**Initial state** $s_0$   The process begins at the initial state when receiving the input task/question.

**Final States** $F$   A set of final states when the process is terminated, which is a subset of $S$.

**Output** $\Gamma$   We define $\Gamma$ to be an infinite set of messages (unit of text) consisting of prompts $P$, language model responses $C$, and feedback from tool/environment $O$: $\Gamma = \{P, C, O\}$, which represents all possible messages that can be generated within `StateFlow`. We further define context history to be $\Gamma^*$, which is a list of messages that have been generated. $S \times \Gamma^*$ could be view as a snapshot of a running `StateFlow` (Rich et al., 2008). Here we distinguish static prompts $P$ that will added to the context directly from instructions $T$ that are used upon calling an LM.

**State transition** $\delta$   Based on the current state and context history, the transition function would determine which state to go to ($\delta : S \times \Gamma^* \to S$). This could mean string matching of

the last input, for example, checking if 'Error' is in the message from code execution. We can also explicitly employ an LLM to check for conditions and determine what is the next state.

**Output Functions** $\Omega$ Here we define $\Omega$ to be a set of output functions, where a function $\omega$ takes the whole context history and generates an output ($\omega : \Gamma^* \to \Gamma$). The output function can be an LLM, a tool call, or a prompter function that returns a static prompt (e.g., P, E and $M_i$ in Figure 1). The generated response will then be added to the context history.

The process starts at $s_0$ when task $Q$ is appended to the context history $\Gamma^*$ and ends when reaching one of the final states (See Algorithm 1). We also use a counter that defines maximum turns of transitions to prevent infinite loops. The process returns the exit state and the whole context history in the end.

---

**Algorithm 1** StateFlow

**Require:** task Q, max transitions M, model $\langle S, s_0, F, \delta, \Gamma, \Omega \rangle$, we define *s.outputs* to be list of functions $[\omega_1, \ldots, \omega_i]$, $\omega_i \in \Omega$ for each $s \in S$
1: $\Gamma^* \leftarrow Q$
2: Counter $\leftarrow 0$
3: $s \leftarrow s_0$
4: **while** $s \notin F$ **do**
5:     **for** $\omega$ **in** s.outputs **do**
6:         **do** $\Gamma^* \leftarrow \Gamma^* + \omega(\Gamma^*)$
7:     **end for**
8:     $s \leftarrow \delta(s, \Gamma^*)$
9:     **return** $s, \Gamma^*$ **if** Counter $\geq$ M
10: **end while**
11: **return** $s, \Gamma^*$

---

### 3.2 Deployment Guideline

In this section, we provide guidelines for deploying StateFlow models, with reference to the case study discussed in Section 3.3. In general, StateFlow is designed for tasks that require a designated process to solve. Essentially, creating a state machine involves transforming abstract control flow or human reasoning into a formalized logical model, grounded in a comprehensive understanding of the task at hand.

**Defining States** To define the states, we start with identifying an ideal workflow of a given task. A state should represent a distinct phase or step in the process, defined with enough granularity to capture key milestones and decision points. With the basic workflow in mind, we need to think about possible situations during the process. Specifically, handling failures is an important part of the state machine, where a Error state is commonly used to handle failures. A set of fine-grained states might lead to better control over a problem-solving process. For example, it is possible to identify different types of errors and use different ways to handle them, but this adds complexity in defining the model as a trade-off.

**Defining output functions** Within each state, we need to define a set of outputs. In practice, we need to identify the set of tools we will use, and what instructions we should send to the LLM. For example, in the Solve state defined for the bash task, we first send an instruction that asks for a bash command, call the model, and then execute the bash command. This is a typical sequence of sending instructions to LLMs and then utilizing tools.

**Defining Transitions** We identify two possible state transitions: 1. A static string matching with the LLM responses or tool executions. For example, the tool execution might return a specific string like "execution failed", which can be used to determine the transition. Also, it is common to instruct LLMs to follow a certain template to generate outputs, so we can extract strings from the response. 2. Use LLM for an explicit checking of the current status when previous context is too random to be used for string matching. For example, we can send the context history to a model to ask whether a given problem is solved.

### 3.3 A Case Study of StateFlow Design on SQL

We use the SQL task from the InterCode (Yang et al., 2023a) benchmark to illustrate how to identify a workflow and convert it to a StateFlow model. The benchmark is evaluated with ReAct (Yao et al., 2022b), a few-shot prompting that requires annotations of example trajectories of actions, thoughts, and observations. These trajectories are human-crafted, and they reflect how one thinks is a good process to solve the problem.

See Figure 2 for one ReAct trajectory for the SQL task: **(1)** the process starts with a 'SELECT' query and results in an error. **(2)** At an error, the next thought is to explore the tables, so the SHOW TABLES command is executed to retrieve all tables. **(3)** After getting the tables, the next step is to explore the 'highschoolers' table with the 'DESC' command. **(4)** With knowing what the table contains, the next thought is to use the select query to solve the question. **(5)** Finally, it confirms the output contains relevant info, and submits. The trajectory demonstrates what to do based on previous history, which is similar to state transitions in `StateFlow`. While the trajectory starts with a 'SELECT' query but results in an error, we believe a better workflow would be to first explore the tables and use the 'SELECT' command. Based on this, we construct the states to be: `Init -> Observe -> Solve -> Verify ->`

> Question: What are the names and grades for each high schooler?
>
> **Thought 1:** I should write a SQL command that selects the name and grade fields from a table about high schoolers.
> **Action 1:** execute[SELECT name, grade FROM high_schoolers]
> **Observation 1:** Error: Table 'network_1.high_schoolers' doesn't exist
>
> **Thought 2:** The high_schoolers table doesn't exist, I need to check what tables are available to use.
> **Action 2:** execute[SHOW TABLES]
> **Observation 2:** [('friend',), ('highschooler',), ('likes',)]
>
> **Thought 3:** There is a table called `highschoolers`. I should check what information this table contains.
> **Action 3:** execute[DESC highschooler]
> **Observation 3:** [('ID', ...), ('name', ...), ('grade', ...)]
>
> **Thought 4:** I can now write a SQL command that answers the question.
> **Action 4:** execute[SELECT name, grade FROM highschooler]
> **Observation 4:** [('John', 12), ('Haley', 10), ...]
>
> **Thought 5:** The output contains the name and grade, answering the question. I can submit this as the answer.
> **Action 5:** submit

Figure 2: A ReAct few-shot example for the SQL task. From the example, we can abstract a general workflow to solve the problem.

`End` (See Figure 1). In each state, the model is instructed to perform a specific action. For example, we ask the model to submit if the task is verified in state `Verify` and explore tables at an error in state `error`.

## 4 Experiments

### 4.1 InterCode Benchmark

We first experiment with 2 tasks from the InterCode benchmark (Yang et al., 2023a): (1) SQL: The InterCode-SQL adapts the Spider dataset for MySQL, containing 1034 task instances. For each task, a MySQL interpreter is set up with all relevant tables within a docker container. (2) Bash: The InterCode-Bash dataset has 200 task instances curated from the NL2Bash dataset. We use the same hyperparameters Zhang et al. (2023a;b; 2024b); Zheng et al. (2023) for both two benchmarks. Specifically, we allow a max of 10 rounds of interaction with the environment. We evaluate with OpenAI GPT-3.5-Turbo and GPT-4-Turbo (both with the 1106 version) and the temperature is set to 0.

**Baselines.** We compare `StateFlow` with two prompting strategies used in the InterCode benchmark. (1) **Plan & Solve** (Wang et al., 2023b): A two-step prompting strategy to first ask the model to propose a plan and then execute it. (2) **ReAct** (Yao et al., 2022b): a few-shot prompting method that prompts the model to generate thoughts and actions. Additionally, since we observed an ideal workflow for the SQL task in Section 3.3, we edit the ReAct prompt used in the benchmark accordingly to see if it already improves performance, named **ReAct_Refined**, and evaluate it on the SQL task (See Table 20 and 21 for details).

**Metrics.** We present metrics provided by the benchmark and we also report the LLM usage of each method. *Success Rate (SR):* a task is considered a success only if the reward is 1. *Error Rate:* percentage of commands that fails. *Turns:* number of interactions with the environment. *Cost:* the cost of calling an LLM API in US dollars.

**StateFlow Setup.** For both tasks, we prompt the model to generate thought and action at each turn. Each prompt consists of 3 components: (1) instruction: details of what the LLM should perform at the current state; (2) examples: partial thought or action steps from the ReAct examples as demonstrations; (3) response format: explain the thought-action template. These prompts are put in the system message of each LLM agent and are not visible to other agents. **SQL:** We construct 6 states for the SQL task (See Figure 1 and Section 3.3 for a case study). In `Init`, we always execute the 'SHOW TABLES' command. In state `Observe`, `Solve`, `Verify`, `Error`, when the execution output is an error string, we will transit to state `Error`. In any of states `Solve`, `Verify`, `Error`, a successful 'DESC' will

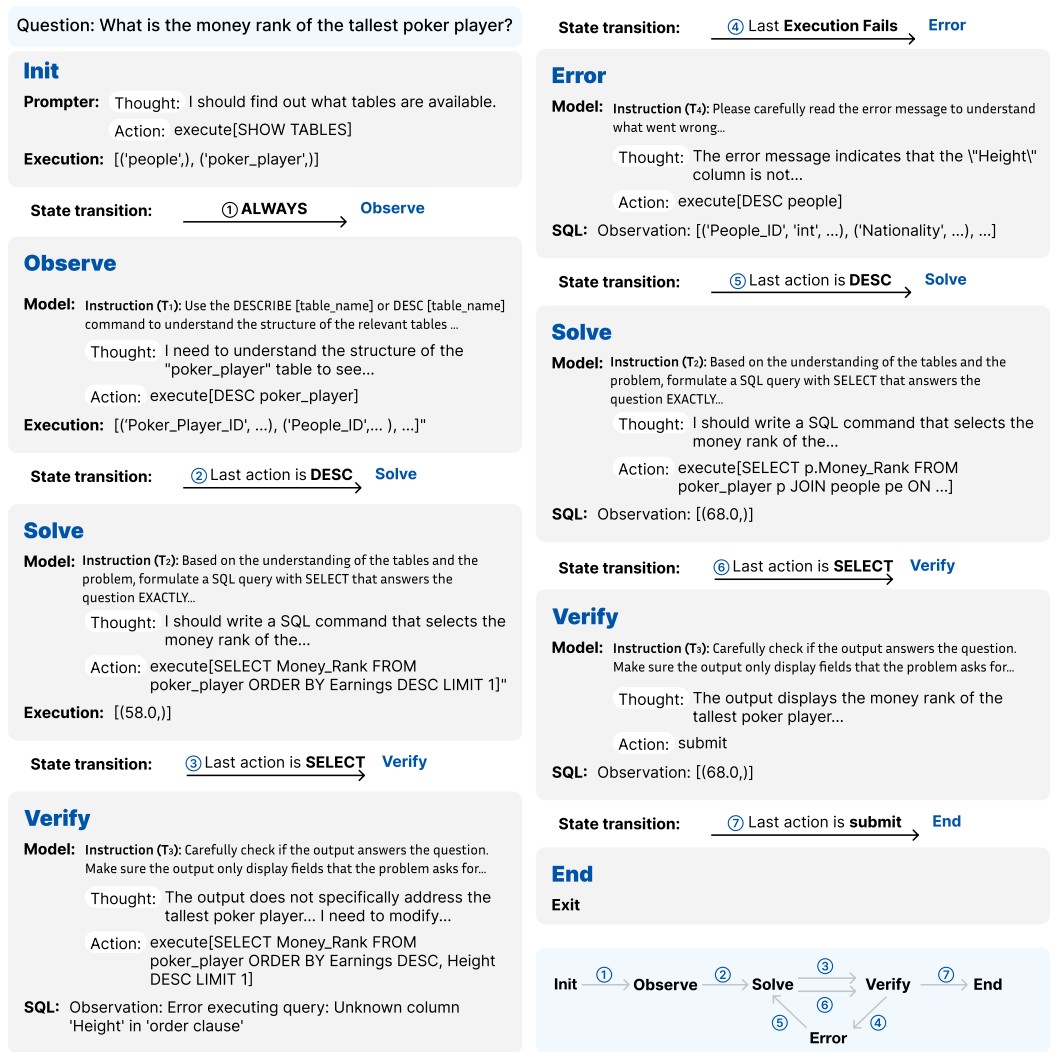

Figure 3: An example of the `StateFlow` execution for the SQL task. In this example, `StateFlow` runs through all states to reach a final answer.

| | GPT-3.5 | | | | GPT-4 | | | |
|---|---|---|---|---|---|---|---|---|
| | SR↑ | Turns↓ | Error↓ | Cost↓ | SR↑ | Turns↓ | Error↓ | Cost↓ |
| Plan & Solve | 47.68 | **4.31** | 12.5 | **2.38** | 56.19 | 5.39 | **1.79** | 44.7 |
| ReAct | 50.68 | 5.58 | 16.3 | 17.7 | 60.16 | 5.26 | 3.87 | 147 |
| ReAct_Refine | 57.74 | 5.47 | **3.82** | 18.1 | 57.93 | **5.01** | 2.49 | 141 |
| **StateFlow** | **63.73** | 5.67 | 6.82 | 3.82 | **69.34** | 5.11 | 1.89 | **36.0** |

Table 1: Evaluation of the Intercode SQL dataset with GPT-3.5 and GPT-4. Best metrics of each model is in **Bold**. Second-best is Underlined.

transit to `Solve`; a successful 'SELECT' will transit to `Verify`. In `Verify`, we use LLM to self-evaluate, which is proven useful by Weng et al. (2022); Xie et al. (2023). **Bash:** For the bash task, we define a `StateFlow` model consists of 5 states `Init, Solve, Verify, End, Error`. The states are similar to SQL, and the transition only depends on whether the execution is successful or not (Figure 1). See details in Appendix A.1

| | GPT-3.5 | | | | GPT-4 | | | |
| --- | --- | --- | --- | --- | --- | --- | --- | --- |
| | SR↑ | Turns↓ | Error↓ | Cost↓ | SR↑ | Turns↓ | Error↓ | Cost↓ |
| Plan & Solve | 23.5 | 4.98 | 25.8 | 0.74 | 20.5 | 5.15 | 21.0 | 9.59 |
| ReAct | 32.5 | 5.52 | 13.2 | 3.28 | 31.5 | 3.86 | 9.90 | 20.40 |
| **StateFlow** | **36.0** | **3.90** | **8.74** | **0.63** | **39.0** | **2.95** | **7.85** | **5.02** |

Table 2: Evaluation of the InterCode Bash dataset with GPT-3.5 and GPT-4. Best metrics of each model is in **Bold**. Second-best is Underlined.

**Result and analysis on SQL**    On GPT-3.5, our refined ReAct version increases the success rate by 7% over the original ReAct prompt. But `StateFlow` can further improve over the refined version by 6%, with 5× less cost. We note that the big difference in cost mainly comes from the prompt token use. The ReAct prompt has 2043 tokens with 4 example trajectories, while the longest instruction from `StateFlow` has only 400 tokens. The LLM is called iteratively with the whole prompt, thus the difference in cost accumulates. Compared to Plan & Solve, `StateFlow` uses 1.6× more cost but improves the SR by 17%. ReAct_Refined and our methods follow a similar workflow and have low error rates. On GPT-4-Turbo, `StateFlow` can improve over ReAct by 10% in SR but has 3× less cost. We further look into results from different levels of difficulties and found that the success rate drops significantly for harder tasks (Details in Appendix A.2). `StateFlow` also shows a greater improvement over ReAct on hard and extra hard tasks, which often require complex joins across multiple tables. We found that state `Error` in `StateFlow` is crucial, as it is encountered in 20% of extra hard tasks compared to only 9% in easy tasks, highlighting its role in improving performance on challenging queries.

**Result and analysis on Bash.**    On the bash task, `StateFlow` outperforms other methods while efficiently interacting with the environment. Switching to GPT-4-Turbo has little effect on the methods, where the two baselines even suffer from a decrement in accuracy. While the success rate is low, the average reward is high as 0.8 (See Table 7 in Appendix). Our investigation shows that 58.5% of the bash tasks have a positive reward greater than 0.5, while only 0.5% of the failed tasks have a positive reward greater than 0 in the SQL task. This is because a bash task sometimes consists of two requests (retrieve information or configure a file), making it harder to completely solve a task. Please see more details and results in Appendix A.2 and A.3.

**Ablation of States.** To understand how different states contribute to the accuracy in `StateFlow`, we perform additional ablations and analysis with the SQL task (See Table 3). **(1)** We remove the `Observe` state. Note that in the original prompt for state `Solve`, we also instruct the model to call 'DESC' if necessary, so the model can still explore tables, but not with an explicit state and instruction to perform this action. **(2)** We remove the error state and rely on the verify state to correct mistakes. **(3)** We remove the verify state and add a sentence in `Solve` to prompt the model to submit when finished. The

| | SR
% ↑ | Turns
↓ | Error
%↓ | Cost
$↓ |
| --- | --- | --- | --- | --- |
| StateFlow | **63.73** | 5.67 | 6.82 | 3.82 |
| No_Verify | 62.28 | **5.18** | **5.96** | **3.68** |
| No_Error | 58.80 | 5.72 | 11.6 | 4.05 |
| No_Obsrve | 57.83 | 6.00 | 17.0 | 4.64 |

Table 3: Ablation of states on the InterCode SQL dataset with GPT-3.5-Turbo. Best metrics in **Bold**. Second-best is Underlined.

table shows that removing any of the states results in a drop in performance. When `Verify` state is removed, `StateFlow` has the lowest cost and error rate, matching the idea that more corrections to the results will be performed with an explicit `Verify` state. Removal of the `Error` state leads to a drop of 5% in SR, and an increase in turns, error rate, and cost, showing that the `Error` state plays an important role in the workflow. Removal of the `Observe` results in the lowest SR and highest cost, showing that 'Observe' is the most important state.

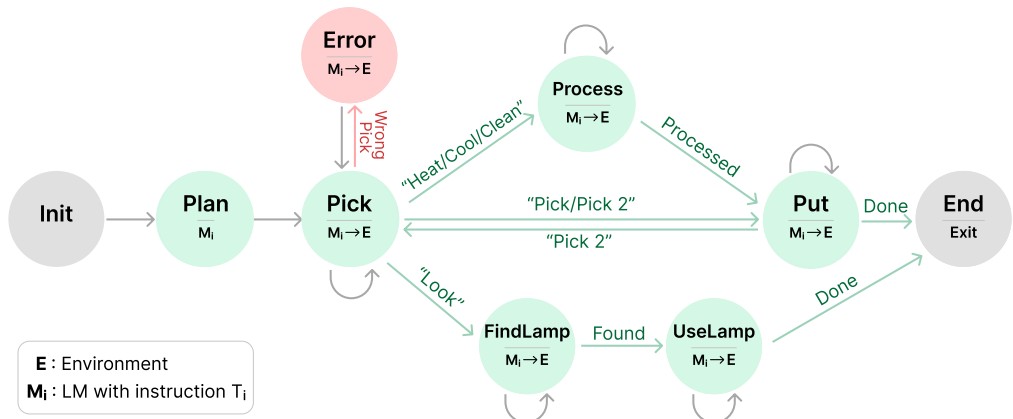

Figure 4: The `StateFlow` model for ALFWorld. For `Plan`, we call the LLM directly. For other states (except `Init` and `End`), we first call LLM with an instruction and then call the environment to get feedback. In state `Pick`, when the correct object is picked, we transit to different states based on task type. For states `Pick`, `Process`, `FindLamp`, `UseLamp`, `Put`, we stay in the current state if the corresponding task is not completed, represented by gray semi-circle arrows.

|  | Pick | Clean | Heat | Cool | Look | Pick 2 | All↑ | Cost \$↓ |
|---|---|---|---|---|---|---|---|---|
| ReAct | 83.3 | 36.6 | 53.6 | 58.7 | 63 | 41.2 | 55.5 | 6.6 |
| ALFChat (2 agents) | 87.5 | 60.2 | 44.9 | 65.1 | 38.9 | 43.1 | 58.2 | 6.9 |
| ALFChat (3 agents) | 84.7 | 60.2 | 69.6 | 77.8 | 68.5 | 41.2 | 67.7 | 6.1 |
| **StateFlow** | **91.7** | **83.9** | **85.5** | **79.4** | **92.6** | **62.7** | **83.3** | **2.6** |

Table 4: Performance and cost of `StateFlow` and other methods on ALFWorld benchmark with GPT-3.5-Turbo. We report average success rate of 3 attempts.

## 4.2 ALFWorld

ALFWorld (Shridhar et al., 2020) is a synthetic text-based game implemented in the TextWorld environments (Côté et al., 2019). It contains 134 tasks across 6 distinct task types: move one or two objects (e.g., put one/two cellphone in sofa), clean/cool/heat an object (e.g., clean/cool/heat some apple and put it in sidetable) and examine an object with lamp (e.g., look at bowl under the desklamp). The agent is required to navigate around a household setting and manipulate objects through text actions (e.g., go to desk 1, take soapbar from toilet 1), and the environment will give textual feedback after each action. We experiment with GPT-3.5-Turbo (1106). For all experiments, we follow AutoGen (Wu et al., 2023a) to use the BLEU metric to map output to the valid action with the highest similarity. Our implementation is also based on AutoGen[1]. More details are in Appendix B.1.

**Baselines.** We evaluate with: 1. **ReAct** (Yao et al., 2022b): We use the two-shot prompt from the ReAct. Note there is a specific prompt for each type of task. 2. **ALFChat (2 agents)** (Wu et al., 2023a): A two-agent system setting from AutoGen consisting of an assistant agent and an executor agent. ALFChat is based on ReAct, which modifies the ReAct prompt to follow a conversational manner. 3. **ALFChat (3 agents)**: Based on the 2-agent system, it introduces a grounding agent to provide commonsense facts whenever the assistant outputs the same action three times in a row.

**StateFlow Setup.** The `StateFlow` model is shown in Figure 4. The process starts at `Plan`, where a plan to solve the task is generated. We note a similar planning is also used in the ReAct prompting. Then, we transit to `Pick` to instruct the model to search for the target object and take it. We stay in `Pick` until the target object is picked.

---

[1] We use AutoGen v0.2.17.

We identify the target object by calling an LLM at the beginning and use it as ground truth. If a wrong object is picked, we go to the `Pick_Error` state, where the wrong object is put down. If the correct object is picked, we transit to the next state (`Process`, `Put` or `FindLamp`) based on the task type. We follow the ReAct template to prompt the model to generate thought and action each time. Also, we use task-specific planning examples and instructions in `Plan` and `Process`. See details in Appendix B.1.

**Results and analysis.** Table 4 shows the results for the ALFWorld benchmark. We record the accuracy for each type of task and also the cost for LLM inference. We can see that `StateFlow` achieves the best performance on all 6 tasks, and significantly outperforms all baseline methods on the whole dataset. It improves over ReAct by 28% and ALFChat (3 agents) by 15%. At the same time, `StateFlow` uses 2.5x less cost. With `StateFlow`, we decompose a long prompt into shorter but more concise prompts to be used when entering a state. Thus, we reduce the prompt tokens used while making the model focus on a sub-task for better responses.

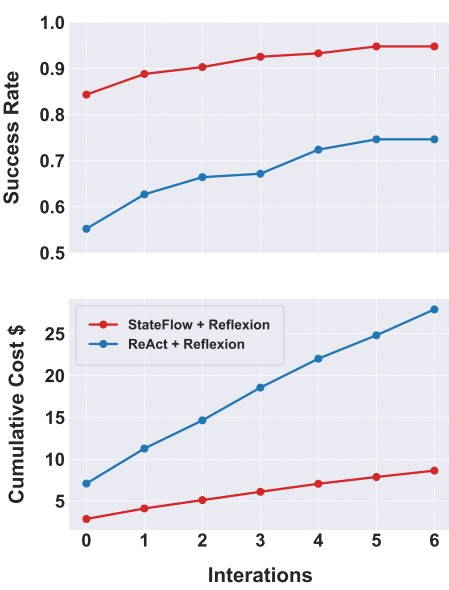

To understand the failure reasons of `StateFlow`, we analyzed all 23 failed tasks and classified them based on the ending state. The analysis revealed that 15 out of 21 tasks ended in the `Pick` state, indicating that finding the correct object is the most challenging part. The failures in the `Pick` state were due to three main reasons: the LLM hallucinating about the object's location, picking the wrong object, and getting stuck in loops between locations. We refer to Appendix B.2 for more analysis of the results. In Appendix B.3, we show that further decomposition of the states in `StateFlow` can increase the success rate to 88.8%, with 15% less cost.

Figure 5: `StateFlow` and ReAct integrated with Reflexion. `StateFlow` can further be improved with Reflexion, with much less cost incurred than ReAct.

**Integration with Reflexion.** We incorporate `StateFlow` with Reflexion (Shinn et al., 2023), showing that `StateFlow` can be combined with iterative refining methods to improve its performance. Reflexion reflects from previous unsuccessful trials and stores the feedback in memory for subsequent trials. We can either use ReAct or `StateFlow` as the basic executor. We run `StateFlow` +Reflexion and ReAct+Reflexion for 6 iterations, until both ReAct and `StateFlow` stop performance improvement. In Figure 5, `StateFlow` +Reflexion further improves from 84% to 94.8%, with the total cost increased from $2.9 to $8.6. While ReAct+Reflexion improves from 55.2% to 74.6%, the total cost for it increased from $7.1 to $27.9.

## 5 Related work

Different prompting frameworks have been proposed to enhance LLM reasoning processes (Zhang et al., 2023d; Wu et al., 2022; Sel et al., 2023; Ning et al., 2023; Zhou et al., 2022; Zhang et al., 2023c; Zelikman et al., 2023). Tree of thoughts (ToTs) (Yao et al., 2023) models the reasoning process as a tree and employs DFS or BFS search to explore sequential thoughts. Tree-of-Thought by (Long, 2023) models the thoughts as trees but relies on a rule-based verifier to determine if a thought is valid and performs refining or backtracking based on a controller. Graph of Thoughts (Besta et al., 2023) models the process as a directed graph and defines 3 types of transformations at a node in the graph: aggregation, refining, and generation. These frameworks have better control over LLM's intermediate steps, but they are not well-designed to consider LLM workflows with external tools and environments.

`StateFlow` considers external feedback and also allows the design of complex patterns for more difficult tasks, where any state can be connected with the definition of state transitions. The step-wise search from ToTs (Yao et al., 2023) can easily be applied to `StateFlow`. When we call LLM in a state, we can generate several responses and employ an evaluator to select the best one.

LLMs have been used to interact with environments (Deng et al., 2023; Yao et al., 2022a; Shridhar et al., 2020) and tools (Paranjape et al., 2023; Gou et al., 2023; Schick et al., 2023; Gao et al., 2023; Yang et al., 2023b; Zhang et al., 2024a; Zou et al., 2024). ReAct (Yao et al., 2022b) uses a few-shot prompting strategy that generates the next action based on past actions, and observations, which have been proven effective. Follow-up works employ iterative refining to improve from previous trials (Sun et al., 2024; Prasad et al., 2023). Reflexion (Shinn et al., 2023) and Self-Refine (Madaan et al., 2023) generate reflections from the past to improve future trials. In parallel to our work, Liu & Shuai (2023) adapts a state machine to record and learn from past trajectories. Extending from ToTs and RAP (Hao et al., 2023), (Zhou et al., 2023) proposes an LLM-based tree search incorporating reflection and feedback from the environment. These methods incur additional costs from the interactions or searching processes. `StateFlow` is orthogonal to these methods, and some of them can be used on top of `StateFlow` to further improve performance.

LLMs are becoming a promising foundation for developing autonomous agents (Xi et al., 2023; Wang et al., 2023a; Wu et al., 2023a; Li et al., 2023; Hong et al., 2023; Sumers et al., 2023). AutoGen (Wu et al., 2023a) offers an open-source platform for developing LLM-based agents. MetaGPT (Hong et al., 2023) proposes a multi-agent framework for software development and CAMEL (Li et al., 2023) proposes a framework for autonomous agent cooperation. In `StateFlow`, specialized agents with different instructions are used in different states.

## 6 Conclusion

In this paper, we propose `StateFlow`, a novel problem-solving framework to use LLMs for complex, multi-step tasks with enhanced efficiency and control. `StateFlow` grounds the progress of task-solving with states and transitions, ensuring clear tracking and management of LLMs' responses and external feedback. We can define a sequence of actions within each state to solve a sub-task. `StateFlow` requires humans to have a good understanding of a given task and build the model and prompts. An intriguing avenue for further research lies in the automation of `StateFlow` model construction and prompting writing, leveraging LLMs to dynamically generate and refine workflows. Further, the idea of employing active learning strategies to iteratively adjust or "train" a `StateFlow`, adding or removing states automatically based on task performance, presents a promising path toward maximizing efficiency and adaptability in complex task solving.

## Acknowledgements

We would like to thank Hanjun Dai and Eric Zelikman for their reviews and helpful feedback. We also thank Yu Tong (Tiffany) Ling from MathGPTPro for her help in creating demonstration figures.

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

# A  InterCode

## A.1  Experiment Details

For ReAct and Plan & Solve, we use the code from InterCode repository[2]. The `StateFlow` models for the ablation study on the InterCode SQL task are in Figure 8 and the full metrics for the ablation study are shown in Table 8. See Figure 9 for a bash example with `StateFlow`. We include also two examples of ReAct and ReAct_Refined in Table 20 and 21 for comparison.

For the InterCode benchmark, we recorded different metrics provided by the benchmark and we also recorded the LLM usage of each method. Additional metrics: (1) Reward: For SQL, the reward is calculated by Intersection over Union (IoU) of the latest execution output generated against the gold output. For Bash, a customized function is used to evaluate the performance against file system modifications and the latest execution output. (2) Token Count: We also recorded the prompt tokens (input), and completion tokens (output) used by each method. See Table 6 and Table 7 for detailed comparisons.

See Table 16 and 17 for the instructions used for SQL `StateFlow` model and Table 18 and 19 for instructions used for Bash. A uniform prompt that introduces the overall environment is put in the system message, and a specific prompt is put in the head of the user message.

Code is available at `https://github.com/yiranwu0/StateFlow`.

## A.2  Additional Analysis

The additional analysis below is based on results with the GPT-3.5-Turbo model.

**SQL.** The SQL dataset consists of different levels of difficulties (See Table 5). The success rate drops with harder tasks, and the difference in SR between easy and extra hard tasks is as great as 50% with `StateFlow`. Harder queries usually pose several constraints on the data and require looking across tables and joining information from them (Extra hard example: "Which distinctive models are produced by maker with the full name General Motors or weighing more than 3500?". Easy example: "Give the city and country of the Alton airport."). However, `StateFlow` greatly improves over other baselines on hard and extra tasks, leading to 20% improvement compared to ReAct. Since harder tasks require information across tables, they are more likely to result in errors. We collected the states traversed for tasks that are solved successfully, and found that only 9% of the easy tasks go over state `Error`, while 20% of the extra hard tasks have state `Error` visited. This indicates that the state `Error` in `StateFlow` plays an important role in the performance gain in hard tasks.

|  | Easy | Medium | Hard | Extra | All |
|---|---|---|---|---|---|
| Plan & Solve | 80.6 | 49.1 | 29.1 | 13.2 | 47.7 |
| ReAct | 72.2 | 57.6 | 35.6 | 15.7 | 50.7 |
| ReAct_Refined | 77.0 | **66.1** | 44.8 | 19.9 | 57.7 |
| **StateFlow** | **87.9** | 62.9 | **59.8** | **36.7** | **63.7** |

Table 5: Success Rate of different level of difficulties on InterCode SQL with GPT-3.5-Turbo.

**Bash** A Bash task consists of one or both of the following two requests: 1. retrieving information that can be acquired through output (e.g, "find files in /workspace directory which are modified 30 days ago") and 2. changing configuration of a file/folder (e.g., "change permissions for all PHP files under the /testbed directory tree to 755") (Yang et al., 2023a). To complete a Bash task, all required commands need to be correct, making it difficult to achieve a reward of 1. We collect the rewards from the failed bash tasks of `StateFlow` and find that 58.5% of the bash tasks have a positive reward greater than 0.5. For SQL, only 0.5% of the failed tasks have a positive reward greater than 0. This shows that

---

[2]https://github.com/princeton-nlp/intercode

| | | SR %↑ | Reward ↑ | Turns ↓ | Error %↓ | Cost $↓ | Average p-token ↓ | Average c-token ↓ |
|---|---|---|---|---|---|---|---|---|
| GPT-3.5 | Plan & Solve | 47.68 | 0.4893 | **4.31** | 12.46 | **2.38** | **1998** | 154 |
| | ReAct | 50.68 | 0.5257 | 5.58 | 16.33 | 17.73 | 16456 | 348 |
| | ReAct_Refined | 57.74 | 0.5928 | 5.47 | **3.82** | 18.05 | 16782 | 340 |
| | Try Again[*] | 56.38 | 0.5762 | 7.62 | 34.73 | 6.61 | 6098 | **145** |
| | **StateFlow** | **63.73** | **0.6637** | 5.67 | 6.82 | 3.82 | 3128 | 281 |
| | **SF_Chat** | 60.83 | 0.6356 | 5.38 | 5.01 | 5.59 | 4965 | 220 |
| GPT-4 | Plan & Solve | 56.19 | 0.5793 | 5.39 | 1.79 | 44.7 | 3065 | 416 |
| | ReAct | 60.16 | 0.6277 | 5.26 | 3.87 | 147 | 12951 | 419 |
| | ReAct_Refined | 57.93 | 0.6104 | 5.01 | 2.49 | 141 | 12377 | 421 |
| | **StateFlow** | 69.34 | 0.7223 | 5.11 | 1.89 | **36.0** | **2700** | **261** |
| | **SF_Chat** | **70.41** | **0.7329** | **4.84** | **1.20** | 49.2 | 3907 | 283 |

Table 6: Results of the Intercode SQL with GPT-3.5 and GPT-4 with all metrics. We also include SF_Chat, an alternative of StateFlow, and another baseline Try Again with an oracle setting([*]). Best metrics of each model is in **Bold**. Second-best is Underlined.

| | | SR %↑ | Reward ↑ | Turns ↓ | Error %↓ | Cost $↓ | Average p-token ↓ | Average c-token ↓ |
|---|---|---|---|---|---|---|---|---|
| GPT-3.5 | ReAct | 32.5 | 0.7674 | 5.52 | 13.23 | 3.28 | 15529 | 442 |
| | Plan & Solve | 23.5 | 0.7472 | 4.98 | 25.78 | 0.74 | 3232 | 225 |
| | Try Again[*] | **49.5** | **0.8453** | 6.88 | 19.54 | 0.83 | 3833 | 159 |
| | **StateFlow** | 36.0 | 0.8033 | 3.90 | **8.74** | **0.63** | **2667** | 232 |
| | **SF_Chat** | 37.0 | 0.8011 | **3.04** | 9.95 | 0.79 | 3658 | **148** |
| GPT-4 | ReAct | 31.5 | 0.7724 | 3.86 | 9.90 | 20.40 | 9027 | 392 |
| | Plan & Solve | 20.5 | 0.7280 | 5.15 | 21.03 | 9.59 | 3460 | 444 |
| | **StateFlow** | **39.0** | **0.8059** | 2.95 | 7.85 | **5.02** | **1835** | 225 |
| | **SF_Chat** | 37.5 | 0.8015 | **2.86** | **7.27** | 7.04 | 3113 | **135** |

Table 7: Results of the Intercode Bash with GPT-3.5 and GPT-4 with all metrics. We also include SF_Chat, an alternative of StateFlow, and another baseline Try Again with an oracle setting([*]).

many of the tasks are partly solved. In Section A.3, the results of "Try Again" show that signals from the environment are very useful and help the model understand how much process it has made to solve the task. Currently, our StateFlow model designed for bash follows a simple workflow of solve→verify. However, since the bash task consists of two distinct requests as mentioned, we believe it is possible to improve the performance with a more complex StateFlow model, with different states constructed for each request.

### A.3 Additional Results

**SF_Chat** We present the results of SF_Chat, an alternative version of StateFlow, in Table 6 and 7. Instead of creating individual LLM agents with specific instructions, we directly append that instruction to context history, imitating a user's reply in a conversation. For this alternative, we construct the context history in a conversational manner. The observations and instructions are appended as user messages, and replies from models are appended as assistant messages. We can see that SF_Chat has a similar performance to StateFlow. By directly appending the instructions, SF_Chat takes fewer turns to finish the task and has a lower error rate than StateFlow. In trade-off, the cost of SF_Chat is higher. We note that SF_Chat might not be suitable for tasks that require many turns of interactions (e.g., ALFWorld), because the cost would be high with the instruction prompts accumulated in the context history.

| | SR % ↑ | Reward ↑ | Turns ↓ | Error % ↓ | Cost $ ↓ | Average p-token ↓ | Average c-token ↓ |
|---|---|---|---|---|---|---|---|
| StateFlow | **63.73** | **0.6637** | 5.67 | 6.82 | 3.82 | 3128 | 281 |
| No_Verify | 62.28 | 0.6473 | **5.18** | **5.96** | **3.68** | **3070** | **244** |
| No_Error | 58.80 | 0.6091 | 5.72 | 11.58 | 4.05 | 3280 | 318 |
| No_Observe | 57.83 | 0.6041 | 6.00 | 16.95 | 4.64 | 3816 | 337 |

Table 8: Ablation of states on the SQL dataset with GPT3.5-Turbo with all metrics. Best metrics in **Bold**. Second-best is Underlined.

**Try Again** We also include results of another baseline "Try Again" from InterCode Benchmark with GPT-3.5-Turbo. Try Again is an iterative feedback setup from InterCode to mimic human software development (Yang et al., 2023a). In this setup, the model can receive a ground-truth reward from the environment at each execution of the command and stops when the task is solved correctly or reaches max turns. Then the max reward from all the executions is retrieved. We note that this is an **oracle setting** not used in StateFlow and other baselines. In our setting, we use the model to determine when to stop and submit the answer, and only the result before submission is evaluated. From Table 6, we can see that Try Again doesn't work well in the SQL task, and that the performance is slightly worse than our refined version of ReAct. It also has a high error rate of 34.73%. However, with the bash task, Try Again significantly outperforms other methods. This discrepancy indicates the difference in the nature of the two tasks. In the SQL task, the reward is mostly 0 or 1. The observation commands such as "DESC" and a wrong "SELECT" command receive 0, and there are only a few cases where the "SELECT" command is partially correct to receive a partial reward. In this case, the reward signal is not very useful. However, the bash tasks from InterCode are explicitly selected with utilities $\geq 4$ (require several commands), and each correct command can receive a partial reward. Thus, the reward signal can help the model understand how much progress it has made and provide guidance, leading to a significant improvement in performance.

## B  ALFWorld

### B.1  Experiment Details

For ALFWorld, we use ReAct from Reflexion[3], which has the same implementation as the original ReAct repository. For ALFChat, we used the code from AutoGen Evaluation[4]. We allow a maximum of 50 rounds of interactions with the environment. For ReAct and StateFlow, we follow a text completion manner to use the chat-based model GPT-3.5-Turbo. In this experiment, we follow Reflexion to put all instructions and interaction history in one user message when querying the model. The ALFChat is essentially a chat version of ReAct, where the examples are converted into a history conversation between the 'user' and 'assistant'.

**Details on StateFlow Setup** We refer to Table 13, 14, 15 for the prompts we used. The type of input task, as a prior knowledge, is used in all methods. ReAct and ALFChat use different few-shot examples for different types of tasks. Similarly, we have different planning examples for different tasks, as shown in Table 13. In Process, we also have three different prompts corresponding to heat, cool, and clean. In Figure 7, we illustrate activated states for different tasks. As discussed in the main paper, we identify the object of interest by calling the same LLM at the beginning of the task. In Pick, only when we detect a string match of "You pick up A" (A is the object needed), we would transit the next state. Similarly, in FindLamp, we transit to UseLamp only if a string "desklamp" is matched. In Process, we match the pattern "You heat/cool/clean" to proceed. Note that this feedback is from the

---

[3]https://github.com/noahshinn/reflexion
[4]https://github.com/qingyun-wu/autogen-eval

environment. Finally at state `Put` and `UseLamp`, we transit to `End` only if the task succeeds or fails. A task is considered success if we receive "Done=True" from environment, and considered fail if the same response is generated by the LLM for three consecutive rounds (following AutoGen). In this experiment, the environment feedback "Done" is available to all methods to terminate the process upon success. Please see Figure 10 for an example.

## B.2    Additional Analysis

To understand the failure reasons of `StateFlow`, we manually went through the 23 failed tasks of one attempt and classified them based on the ending state (See Table 9). Ending in different states can show how much progress has been made on a task. For example, a task ending in `Cool` implies that the correct object is being picked, but not cooled correctly. From the table, we can see that 15/21 tasks end in `Pick`. This suggests that the most difficult part is to go around the household to find the correct object. For the failed tasks that ended in `Pick`, we summarize three failure reasons: 1. The LLM hallucinates about taking the target object from locations where it does not exist. 2. The LLM takes the wrong object. 3. The LLM gets stuck in loops between two locations.

| Ending State | Pick/Find | Put | Cool | Heat | FindLamp | Error | All |
|---|---|---|---|---|---|---|---|
| StateFlow (7-state) | 15 | 2 | 2 | 1 | 1 | 0 | 21 |
| StateFlow (10-state) | 8 | 3 | 0 | 0 | 0 | 2 | 13 |

Table 9: Count of failed ALFWorld tasks that end in different states. We also include the `StateFlow` model with 10 states for comparison.

## B.3    Additional Experiments

**Adding more states to `StateFlow`**   In the states defined for ALFWorld, we allow different types of actions to be performed. For example, in `Pick`, the model is instructed to either go around the household with the 'go to {recept}' command, open receptacles with 'open', and take an object with 'take {obj} from {recept}' command (the format annotation is adopted from Prasad et al. (2023)). We further divide these actions and add 3 more states and test its performance (See Figure 6). In Figure 10, we show the results of `StateFlow`, and `StateFlow` with 3 more states. The overall performance increases from 83.3% to 88.8%, with 15% less cost. Table 9 indicates that the primary contribution to performance improvement comes from dividing the `Pick` action into `Find` and `Take`.

| | Pick | Clean | Heat | Cool | Look | Pick 2 | All | Cost $ |
|---|---|---|---|---|---|---|---|---|
| StateFlow (7-state) | 91.7 | 83.9 | 85.5 | 79.4 | **92.6** | **62.7** | 83.3 | 2.6 |
| StateFlow (10-state) | **100** | **92.5** | **94.2** | **87.3** | 90.7 | 58.8 | **88.8** | **2.2** |

Table 10: Performance and cost of `StateFlow` and `StateFlow` with 3 more states. We report an average success rate of 3 attempts. More states and division of tasks can further improve performance at even a lower cost.

**Additional results with GPT-3.5-Instruct.**   In our experiments, we use the latest chat models (e.g., GPT-3.5-Turbo) because they are more powerful and have been studied extensively lately. Original ReAct was tested on completion model text-davicin-002, which has been depreciated. The recommended replacement is GPT-3.5-instruct[5]. We also tested ReAct and `StateFlow` on GPT-3.5-turbo-instruct, and find that the performance drops for both methods (see Figure 12). Compared to results on GPT-3.5-Turbo, ReAct has a drop of 4% and `StateFlow` has a drop of 9%.

**Addition results with open-source models**   We further test `StateFlow` and ALFChat with 3 agents on 3 open-source models: Mistral-7b, Llama-8b and Llama3-70b. We observe that

---

[5]https://platform.openai.com/docs/deprecations

|  | ALFChat (3 agents) | | StateFlow | |
|---|---|---|---|---|
|  | Acc (%) | Cost ($) | Acc (%) | Cost ($) |
| Mistral-7b | 1.5 | **0.708** | **6** | 2.03 |
| Llama3-8b | 50.7 | 0.52 | **62.7** | **0.38** |
| Llama3-70b | 83.6 | 2.53 | **94** | **1.83** |

Table 11: Accuracy and cost comparison on ALFWorld with different open-source models. For this experiment, we compare with the strongest baseline and run the test one time.

|  | Pick | Clean | Heat | Cool | Look | Pick 2 | All | Cost $ |
|---|---|---|---|---|---|---|---|---|
| ReAct | 62.5 | 41.9 | 78.3 | **57.1** | 38.9 | 23.5 | 51.5 | 10 |
| StateFlow | **87.5** | **66.7** | **87** | **57.1** | **77.8** | **64.7** | **73.9** | **5.0** |

Table 12: Performance and cost on ALFWorld with GPT-3.5-instruct. Here we report success rate of only one attempt. The performances of both methods drop compared to result with GPT-3.5-Turbo, on which ReAct achieves 55.5% and StateFlow achieves 83.3% in overall success rate.

StateFlow outperforms 3-agent ALFChat with 3 agents across the open-source models. It is worth noting that Mistral-7b struggles to complete the task using both methods, suggesting the model itself is not equipped to handle tasks within ALFWorld. However, Llama3-70b with StateFlow can hit a 94% accuracy, surpassing the performance of GPT-3.5 Turbo.

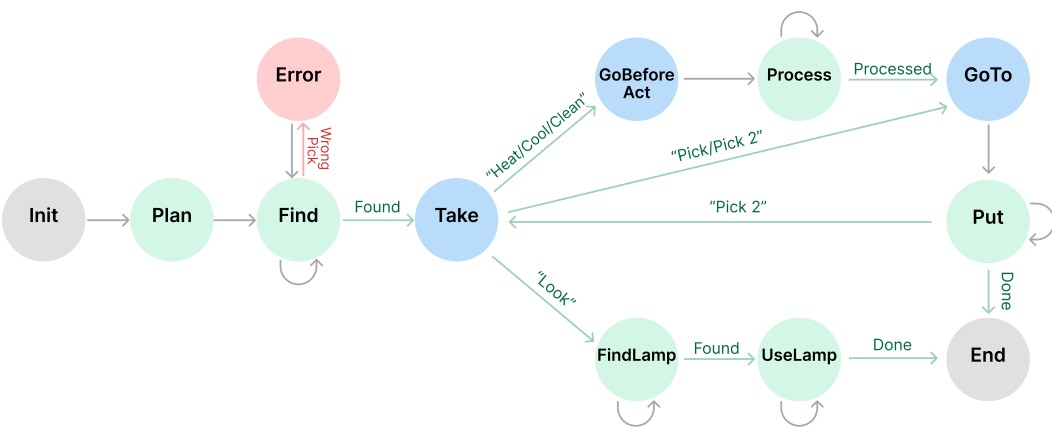

Figure 6: StateFlow model for ALFWorld with 3 addition states. The additional states are marked blue. In states Pick, Process and Put of the original model, there are actually two actions to be performed: 1. go to some receptacle. 2. take/process/put an object. Here, we further split it, using one state for one action.

Figure 7: Active states for different type of tasks in ALFWorld. Only 4-6 states are active for a single task. States `Plan, Pick, Error` and `Put` are sharable across tasks.

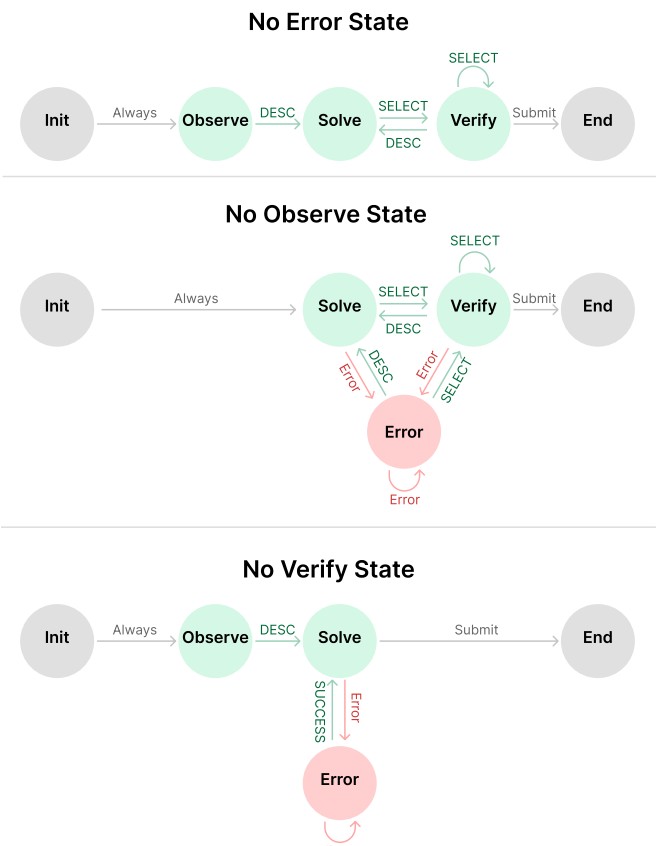

Figure 8: `StateFlow` model for the ablation study with InterCode SQL task.

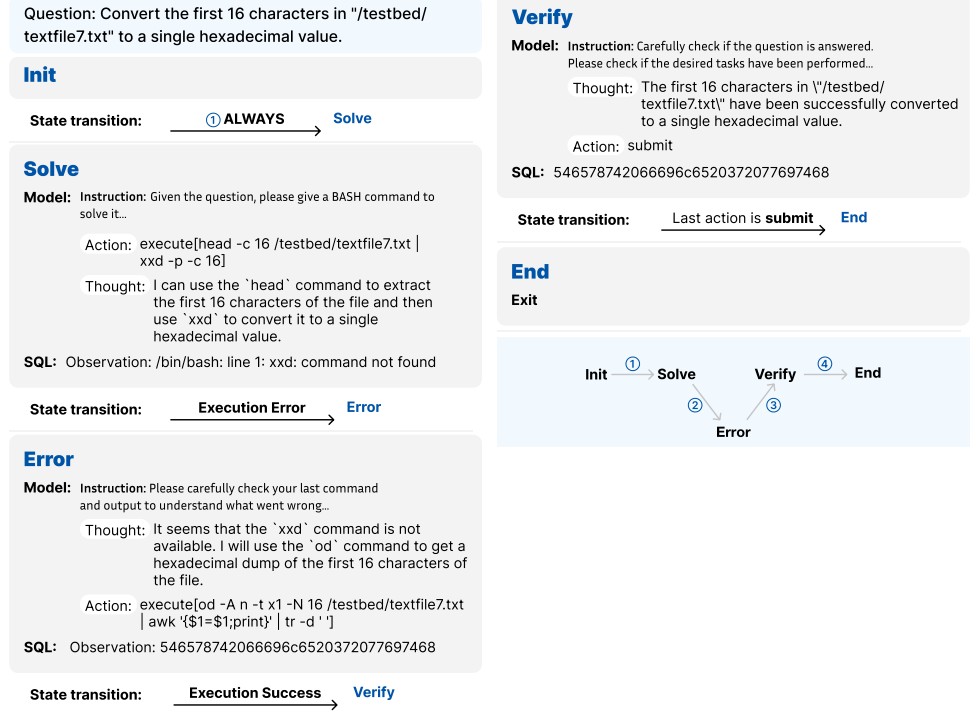

Figure 9: Example of `StateFlow` on Bash task.

Household setting: You are in the middle of a room. Looking quickly around you, you see a cabinet 6, a cabinet 5, a cabinet 4, a cabinet 3, a cabinet 2, a cabinet 1, a coffeemachine 1, a countertop 3, a countertop 2, a countertop 1, a drawer 3, a drawer 2, a drawer 1, a fridge 1, a garbagecan 1, a microwave 1, a shelf 3, a shelf 2, a shelf 1, a sinkbasin 1, a stoveburner 4, a stoveburner 3, a stoveburner 2, a stoveburner 1, and a toaster 1.
Your task is to: heat some apple and put it in garbagecan.

### Init

State transition: ————— Always ————→ **Plan**

### Plan

**Model:** Instruction: You are given a household setting and a task to accomplish. Please make a plan to complete the task ...

Plan: I need to 1. search around for an apple and take it, 2. go to microwave and heat it, 3. go to garbagecan and put it down. An apple is more likely to appear in fridge (1), countertop (1-3), diningtable (1), drawer (1), cabinet (1-6), garbagecan (1). I can check one by one, starting with fridge (1).

State transition: ————— ① Always ————→ **Pick**

### Pick

**Model:** Instruction: Please follow the plan to check receptacles in the household one by one to find the object of interest....

Action: go to fridge 1

**ALFWorld:** Observation: The fridge 1 is closed.

State transition: ————— ② No Pick ————→ **Pick**

### Pick

**Model:** Instruction: Please follow the plan to check receptacles in the household one by one to find the object of interest....

Action: open fridge 1

**ALFWorld:** Observation: You open the fridge 1. The fridge 1 is open. In it, you see a bowl 1, a pan 1, a plate 1, and a tomato 1.

State transition: ————— ③ No Pick ————→ **Pick**

### Pick

**Model:** Instruction: Please follow the plan to check receptacles in the household one by one to find the object of interest....

Thought: I didn't find the apple in the fridge. I should check the next possible location.

Action: go to countertop 1

**ALFworld:** Observation: On countertop 1, you see a apple 1, a egg 1, a mug 2, a soapbottle 2, a spoon 3, and a vase 2.

State transition: ————— ④ No Pick ————→ **Pick**

### Pick

**Model:** Instruction: Please follow the plan to check receptacles in the household one by one to find the object of interest....

Thought: Thought: I found the apple (1) on the countertop. I need to take it.

Action: take apple 1 from countertop 1

**ALFWorld:** Observation: You pick up apple 1 from countertop 1.

State transition: ————— ⑤ Correct object picked ————→ **Solve**

### Heat

**Model:** Instruction: You now take the object of interest with you. Now, please go to the microwave to heat the object...

Action: go to microwave 1

**ALFWorld:** Observation: The microwave 1 is closed.

State transition: ————— ⑥ Object Not heated ————→ **Heat**

### Heat

**Model:** Instruction: You now take the object of interest with you. Now, please go to the microwave to heat the object...

Action: open microwave 1

**ALFWorld:** Observation: You open the microwave 1. The microwave 1 is open. In it, you see nothing.

State transition: ————— ⑦ Object Not heated ————→ **Heat**

### Heat

**Model:** Instruction: You now take the object of interest with you. Now, please go to the microwave to heat the object...

Action: heat apple 1 with microwave 1

**ALFWorld:** Observation: You heat the apple 1 using the microwave 1.

State transition: ————— ⑧ Object heated ————→ **Put**

### Put

**Model:** Instruction: You now take the object of interest with you. Now, please go to the required place to put down the object.

Action: go to garbagecan 1

**ALFWorld:** Observation: On the garbagecan 1, you see a papertowelroll 1, and a potato 1.

State transition: ————— ⑨ Task not done ————→ **Put**

### Put

**Model:** Instruction: You now take the object of interest with you. Now, please go to the required place to put down the object.

Action: put apple 1 in garbagecan 1

**ALFWorld:** Task success

State transition: ————— ⑩ Task success ————→ **End**

### End

**Exit**

Figure 10: Example of `StateFlow` on ALFWorld.

**pick_and_place**   ## Examples
Your task is to: put some spraybottle on toilet.
Plan: I need to 1. search around for spraybottle and take it. 2. go to toilet and put it down. A spraybottle is more likely to appear in cabinet (1-4), countertop (1), toilet (1), sinkbasin (1-2), garbagecan (1). I can check one by one, starting with cabinet 1.
Your task is to: find some apple and put it in sidetable.
Plan: I need to 1. search around for an apple and take it. 2. go to sidetable and put it down. An apple is more likely to appear in fridges (1), diningtables (1-3), sidetables (1), countertops (1), sinkbasins (1), garbagecan (1). I can check one by one, starting with fridge 1.

---

**pick_clean_then_place**
You must use the sinkbasin to clean the object.
## Examples
Your task is to: put a clean lettuce in diningtable.
Plan: I need to 1. search around for some lettuce and take it, 2. go to a sinkbasin and clean it, 3. go to diningtable and put it down. First I need to find a lettuce. A lettuce is more likely to appear in fridge (1), diningtable (1), sinkbasin (1), stoveburner (1-3), cabinet (1-13). I can check one by one, starting with fridge 1.
Your task is to: clean some apple and put it in sidetable.
Plan: I need to 1. search around for some apple and take it, 2. go to a sinkbasin and clean it, 3. go to sidetable and put it down. First I need to find an apple. An apple is more likely to appear in fridges (1), diningtable (1-3), sidetable (1), countertop (1), sinkbasin (1), garbagecan (1). I can check one by one, starting with fridge 1.

---

**pick_heat_then_place**   ## Examples
Your task is to: heat some egg and put it in diningtable.
Plan: I need to 1. search around for an egg and take it, 2. go to microwave and heat it, 3. go to diningtable and put it down. An egg is more likely to appear in fridge (1), countertop (1-3), diningtable (1), stoveburner (1-4), toaster (1), garbagecan (1), cabinet (1-10). I can check one by one, starting with fridge 1.
Your task is to: put a hot apple in fridge.
Plan: I need to 1. search around for an apple and take it, 2. go to microwave and heat it, 3. go to fridge and put it down. An apple is more likely to appear in fridge (1), diningtable (1), coffeetable (1), drawer (1), cabinet (1-13), garbagecan (1). I can check one by one, starting with fridge 1.

---

**pick_cool_then_place**   ## Examples
Your task is to: cool some pan and put it in stoveburner.
Plan: I need to 1. search around for a pan and take it, 2. go to fridge and cool it, 3. go to stoveburner and put it down. An pan is more likely to appear in stoveburner (1-4), sinkbasin (1), diningtable (1), countertop (1-2), cabinet (1-16), drawer (1-5). I can check one by one, starting with stoveburner 1.
Your task is to: put a cool mug in shelf.
Plan: I need to 1. search around for a mug and take it, 2. go to fridge and cool it, 3. go to shelf and put it down. A mug is more likely to appear in countertop (1-3), coffeemachine (1), cabinet (1-9), shelf (1-3), drawer (1-9). I can check one by one, starting with countertop 1.

---

**look_at_obj**   ## Examples
Your task is to: look at bowl under the desklamp.
Plan: I need to 1. search around for a bowl and take it, 2. find a desklamp and use it. First I need to find a bowl. A bowl is more likely to appear in drawer (1-3), desk (1), sidetable (1-2), shelf (1-5), garbagecan (1). I can check one by one, starting with drawer 1.
Your task is to: examine the pen with the desklamp.
Plan: I need to 1. search around for a pen and take it, 2. find a desklamp and use it. First I need to find a pen. A pen is more likely to appear in drawer (1-10), shelf (1-9), bed (1), garbagecan (1). I can check one by one, starting with drawer 1.

---

**pick_two_obj**   ## Examples
Your task is to: put two creditcard in dresser.
Plan: I need to 1. search around for a creditcard and take it, 2. go to dresser and put it down. 3. find another creditcard, 4. go to dresser and put it down. First I need to find the first creditcard. A creditcard is more likely to appear in drawer (1-2), coutertop (1), sidetable (1), diningtable (1), armchair (1-2), bed (1). I can check one by one, starting with drawer 1.
Your task is to: put two cellphone in sofa.
Plan: I need to 1. search around for a cellphone and take it, 2. go to sofa and put it down. 3. find another cellphone, 4. go to sofa and put it down. First I need to find the first cellphone. A cellphone is more likely to appear in coffeetable (1), diningtable (1), sidetable (1-2), drawer (1-4), sofa (1), dresser (1), garbagecan (1). I can check one by one, starting with coffeetable 1.

Table 13: Few-shot examples for State Plan in ALFWorld. Same as ReAct and other baselines, we use one prompts for each type of task.

---

**Head of system message**
You are given a description of a household, please interact with the household to solve the task.
This is a simulation of and all the actions are high-level shortcuts. Follow the instructions to give your next reply.

## RESPONSE FORMAT
Reply with the following template (<...> is the field description):
Thought: <your thought>
Action: <your action>
or
Action: <your action>
In you reply, you can give both a thought and an action, or just an action. You can only give one action at a time.

## Environment feedback
After each of your turn, the environment will give you immediate feedback.
Observation: <observation>

---

**Pick**
## Instructions
Please follow the plan to check receptacles in the household one by one to find the object of interest. Each time, you can observe all the objects in the receptacle. Determine if the object you are looking for is in that receptacle.
You need to find the EXACT object that is asked for. For example, if you need to find a "soapbar", only take it when you see a "soapbar {i}" in the receptacle, instead of a "soapbottle {i}".
Use "open {recept}" command to open a receptacle.

## Examples
- Use "go to {recept}" command to go to the receptacle:
Action: go to cabinet 1
Action: go to fridge 1
Action: go to diningtable 1
- Take the object only if the place have the exact object you are looking for:
Thought: Now I find the soapbar (1). Next, I need to take it.
Action: take spraybottle 2 from cabinet 2
Thought: Now I find the apple (1). Next, I need to take it.
Action: take apple 1 from diningtable 1

---

**Error**
You just took the wrong object. Please put it down with the "put obj in/on place" command, where "place" is the place where you took the object from.
Please also give your thought of what is the next place to check based on the plan.

## Examples
Thought: I accidentally took the tomato instead of the apple. I need to put it back, and then check diningtable 2.

Action: put tomato 1 in/on diningtable 1
Thought: The object I want to take is a soapbar, but I took a soapbottle. I need to put it back, and then check the sinkbasin 1.
Action: put soapbottle 4 in/on toilet 1

---

**Process (Heat)**
## Instructions
You now take the object of interest with you. Now, please go to the microwave to heat the object. You must first go to the microwave with the "go to {microwave}" command. Then, use the "heat {obj} with {microwave}" command to heat the object. You don't need to open the microwave or put the object in the microwave.

## Examples:
- If you just picked the object, go to the microwave first:
Action: go to microwave 1
- Then heat the object:
Action: heat apple 1 with microwave 1
Action: heat bread 1 with microwave 1

---

Table 14: Instructions of StateFlow for ALFWorld.

**Process (Cool)**
## Instructions
You now take the object of interest with you. Now, please go to the fridge to cool the object. You must first go to the fridge with the "go to {fridge}" command. Then, use the "cool {obj} with {fridge}" command to cool the object. You don't need to open the fridge or put the object in the fridge.

## Examples:
- If you just picked the object, go to the fridge first:
Action: go to fridge 1
Action: go to fridge 1
- Then cool the object:
Action: cool pan 1 with fridge 1
Action: cool potato 2 with fridge 1

---

**Process (Clean)**
## Instructions
You now take the object of interest with you. Now, please go to the sinkbasin to clean the object. You must first go to the sinkbasin with the "go to {sinkbasin}" command. Then, use the "clean {obj} with {sinkbasin}" command to clean the object. You don't need water or soap to clean the object.

## Examples:
- Go to the sinkbasin first:
Action: go to sinkbasin 1
Action: go to sinkbasin 2
- Then clean the object:
Action: clean lettuce 1 with sinkbasin 1
Action: clean soapbar 4 with sinkbasin 2

---

**Find Lamp**
## Instructions
You have found and taken the object, now please go around to find a desklamp. Plese use "go to {place{" command to go to different places in the household to find a desklamp. Use "open {place}" command to open a closed place if you need to.

## Examples
Action: go to sidetable 1
Action: go to dresser 1

---

**Use Lamp**
## Instructions
You now find a desklamp. Please use the "use {desklamp}" command to look at the object. "{desklamp}" denotes the desklamp you just found. You should not perform any other actions.

## Examples:
1. Observation: On the sidetable 2, you see a desklamp 3, a newspaper 1, and a statue 2.
Action: use desklamp 3
2. On the sidetable 2, you see a alarmclock 1, a desklamp 1, and a pen 2. Thought: Now I find a desklamp (1). Next, I need to use it.
Action: use desklamp 1

---

**Put**
## Instructions
You now take the object of interest with you. Now, please go to the required place to put down the object. You must first go to the receptacle with "go to {recept}" command.
If the receptacle is closed, use the "open {recept}" command to open it. You can only take one object at a time. If you task is to put two objects, please go to the required place to put the first object first. When you are at the place, use the "put obj in/on place" command to put down the object.

## Examples:
- Always go to the receptacle first:
Action: go to sidetable 1
Action: go to toilet 1
Action: go to diningtable 1
- Then put down the object:
Action: put apple 3 in/on sidetable 1
Action: put soapbar 4 in/on toilet 1

---

Table 15: Instructions of `StateFlow` for ALFWorld. (Continued)

**System message**
Interact with a self.setting system using self.language queries to answer a question.

**Observe**
## Instructions
Use the DESCRIBE [table_name] or DESC [table_name] command to understand the structure of the relevant tables. Only give one DESC command in action.

## Examples
Action: execute[DESC highschooler]
Action: execute[DESC friends]

## RESPONSE FORMAT
For action, put your SQL command in the execute[] block.
Reply with the following template (<...> is the field description, replace it with your own response):

Thought: <your thought on which table(s) is/are relevant in one short sentence>
Action: execute[<your command>]

**Error**
## Instructions
Please carefully read the error message to understand what went wrong. If you don't have enough information to solve the question, you can use the DESC [table_name] command to explore another table. You may want to review other tables to see if they have the information you need.

## Examples
Thought: A 'transcripts' table exists, but it doesn't have the 'release_date' column I came up with. I should find out what columns are available.
Thought: The 'friends' table has two ids. I should check if the 'highschooler' table has a name associated with an ID.
Thought: The 'contestants' is a table, it is not a column in 'people'. I need to check the 'contestants' table to see how to get the contestant names.
Thought: I get a single number that is the number of likes that the high schooler Kyle has. This should be the answer.

## RESPONSE FORMAT
For action, put your SQL command in the execute[] block. You should only give one command to execute per turn.
Reply with the following template (<...> is the field description, replace it with your own response):

Thought: <your thought on why this query is error and whether you should gather more information or fix the error in one sentence>
Action: execute[<your command>]

**Verify**
## Instructions Carefully check if the output answers the question exactly. Make sure the output only display fields that the problem asks for. - If the output contains any extra fields, please revise and modify your query (column alias is fine, no need to round numbers). - If the output doesn't answer the question, please revise and modify your query. You may use DESC/DESCRIBE to learn more about the tables. - If the output answers the question exactly, please submit the query with this "Action: submit" command.

## Examples
Thought: The output displays the contestant names and also contestant count. Although the count is used for sorting, it should not be displayed in output. I should modify my query to only select the contestant names.
Thought: The question asks for the total population for North America. However, the output also has the continent id. I should modify my query to only select the total population.

## RESPONSE FORMAT
For action, put your SQL command in the execute[] block. If the problem is solved, your action should be "Action: submit".
Reply with the following template (<...> is the field description, replace it with your own response, "|" is the "or" operation):
Thought: <your thought on whether the output and command answers the problem>
Action: execute[<your new command>] | submit

Table 16: Instructions of `StateFlow` for SQL task (part 1).

**Solve**
## Instructions Based on the understanding of the tables and the problem, formulate a SQL query with SELECT that answers the question EXACTLY. Use specific clauses like WHERE, JOIN, GROUP BY, HAVING, etc if necessary. If you need more information of another table, use DESC to explore the table.
Notes: You should construct your command that the output answers the question exactly. For example, If the question asks for count, your command should output a single number. Only select the field the question asks for. Do not include relevant but unnecessary fields such as ids or counts, unless the question specifically asks for it. No need to CAST or ROUND numbers unless the question asks for it.

## Examples:
Thought: I should write a SQL command that selects the names from a table about high schoolers in ascending order of their grades. Grade should not be selected.
Action: execute[SELECT name, grade FROM high_schoolers ORDER BY high_schoolers.grades ASC]
Thought: I can use the SUM and AVG functions to get the total population and average area values for North America.
Action: execute[execute[SELECT SUM(population) AS total_population, AVG(area) AS avg_area FROM countries WHERE continent = 'North America' AND area > 3000]]
Thought: I should write a SQL query that gets the name field from contestants and exclude the name of 'Jessie Alloway'
Action: execute[SELECT contestant_name FROM contestants WHERE contestant_name != 'Jessie Alloway']
Follow the RESPONSE FORMAT to give your thought and action.

## RESPONSE FORMAT
For action, put your SQL command in the execute[] block.
Reply with the following template (<...> is the field description, replace it with your own response):

Thought: <your thought on constructing command to answer the query exactly>
Action: execute[<your command>]

Table 17: Instructions of `StateFlow` for SQL task (part 2).

**System message**
Interact with a Bourne Shell system using BASH queries to answer a question. Follow the user's instructions to solve the problem.

**Solve**
## Instructions
Given the question, please give a self.language command to solve it.

## Examples
Thought: I can try to use 'od' (octal dump) command to get a hexadecimal dump and stitch together the values into one continuous string.
Action: execute[od -A n -t x1 -N 16 /testbed/textfile7.txt — awk '1 =1;print' — tr -d ' ']
Thought: I should find the paths to all java files in the testbed directory, then apply the word count command to each path.
Action: execute[find /testbed -name "*.java" -type f -exec md5sum  + — sort — uniq -D -w 32 — awk 'print $1']
Thought: I should find the paths to all php files in the testbed directory, then apply the word count command to each path.
Action: execute[find /testbed -name "*.php" -type f -exec cat  + — wc -l]
Thought: The 'du' command is useful for printing out disk usage of a specific directory. I can use the -h option to print in human readable format and the -s option to only print the total disk usage.
Action: execute[du -sh /workspace]

Follow the RESPONSE FORMAT to give your thought and action.
## RESPONSE FORMAT
For action, put your BASH command in the execute[] block. Only give one command per turn.
Reply with the following template (<...> is the field description, replace it with your own response):

Thought: <your thought in one sentence>
Action: execute[<your command>]

Table 18: Instructions of `StateFlow` for Bash task (part 1).

**Error**
## Instruction
Please carefully check your last command and output to understand what went wrong. Revise and modify your command accordingly or try another command.

## Examples
Observation: /bin/bash: line 1: xxd: command not found
Thought: Seems like xxd is not available. I can try to use 'od' (octal dump) command to get a hexadecimal dump.
Action: execute[od -A n -t x1 -N 16 /testbed/textfile7.txt]

Follow the RESPONSE FORMAT to give your thought and action.
## RESPONSE FORMAT
Reply with the following template (<...> is the field description):
Thought: <your thought in one sentence>
Action: execute[<your command>]

**Verify**
## Instructions
Carefully check if the question is answered.
- Please check if the desired tasks have been performed.
- If the question also asks for output, please check your last command and output, and make sure the output is in the desired format, and doesn't contain any extra fields.
- If the desired tasks have been performed, please submit the query with this "Action: submit" command.

## Examples
Thought: This gives me storage information for every folder under the workspace directory, but I only need the storage for just the 'workspace/' directory. The '-s' option should help with this.
Action: execute[du -sh /workspace]
Thought: This shows the output hashes and they have the same values, indicating that these files are duplicates. However, the file names are also shown, which are not needed.
Action: execute[find /testbed -name "*.java" -type f -exec md5sum  + — sort — uniq -D -w 32 — awk 'print $1']
Thought: This shows me too much information, I only need the total number of lines. I should add up the lines together and output a single number.
Action: execute[find /testbed -name "*.php" -type f -exec cat  + — wc -l]
Thought: The hello.txt file has been created successfully in the testbed/ directory, and it contains the Hello World text. I can submit.
Action: submit

Please follow this RESPONSE FORMAT to give your thought and action.
## RESPONSE FORMAT
For action, put your {self.language} command in the execute[] block. Only give one command per turn. If the question is solved, your action should be "Action: submit".
Reply with the following template (<...> is the field description, replace it with your own response, "|" is the "or" operation):

Thought: <your thought on whether the question is answered in one sentence>
Action: execute[<your new command>] — submit

Table 19: Instructions of `StateFlow` for Bash task (part 2).

---

**ReAct Prompt Example 1**

Question: What are the names and grades for each high schooler?

**Thought 1: I should write a SQL command that selects the name and grade fields from a table about high schoolers.**

**Action 1: execute[SELECT name, grade FROM high_schoolers]**

**Observation 1: Error executing query: Table 'network_1.high_schoolers' doesn't exist**

Thought 2: The high_schoolers table doesn't exist, I need to check what tables are available to use.

Action 2: execute[SHOW TABLES]

Observation 2: [('friend',), ('highschooler',), ('likes',)]

Thought 3: There is a table called 'highschoolers'. I should check what information this table contains.

Action 3: execute[DESC highschooler]

Observation 3: [('ID', 'int', 'NO', 'PRI', None, 'auto_increment'), ('name', 'text', 'YES', '', None, ''), ...]

Thought 4: The highschooler table contains the name and grade columns. I can now write a SQL command that answers the question

Action 4: execute[SELECT name, grade FROM highschooler]

Observation 4: [('John', 12), ('Haley', 10), ('Alexis', 11), ('Jordan', 12), ('Austin', 11), ('Tiffany', 9), ('Kris', 10), ('Jessica', 11), ('Jordan', 9), ('Brittany', 10), ('Logan', 12), ('Gabriel', 9), ('Cassandra', 9), ('Andrew', 10), ('Gabriel', 11), ('Kyle', 12)]

Thought 5: The output contains the name and grade for each high schooler, answering the question. I can submit
this as the answer.

Action 5: submit

---

**ReAct_Refined Prompt Example 1**

Question: What are the names and grades for each high schooler?

**Thought 1: I should first find out what tables are available in this MySQL database that can help me answer this question.**

Action 1: execute[SHOW TABLES]

Observation 1: [('friend',), ('highschooler',), ('likes',)]

Thought 2: There is a table called 'highschoolers'. I should check what information this table contains.

Action 2: execute[DESC highschooler]

Observation 2: [('ID', 'int', 'NO', 'PRI', None, 'auto_increment'), ('name', 'text', 'YES', '', None, ''), ...]

Thought 3: The highschooler table contains the name and grade. I can now write a SQL command that answers the question.

Action 3: execute[SELECT name, grade FROM highschooler]

...(Same as original example)

---

Table 20: (Example 1) Comparison of example trajectories from ReAct and refined version of ReAct for the SQL task. For the new version, we always start with the "SHOW TABLES" command. In the original ReAct prompt, it always starts with a select query but results in error.

---

**ReAct Example 2**

Question: What are the names of the contestants whose names are not 'Jessie Alloway'

**Thought 1: I should write a SQL query that gets the name field from contestants and exclude the name of 'Jessie Alloway'**

**Action 1: execute[SELECT name FROM contestants WHERE name != 'Jessie Alloway']**

**Observation 1: Error executing query: Unknown column 'name' in 'field list'**

Thought 2: It looks like there is a table called 'contestants', but it doesn't have a name column. I need to find out what columns are available.

Action 2: execute[DESCRIBE contestants]

Observation 2: [('contestant_number', 'int', 'NO', 'PRI', None, 'auto_increment'), ('contestant_name', 'varchar(50)', 'NO', ...)]

Thought 3: The available columns for the 'contestant' table are 'contestant_number' and 'contestant_name'. I can now write a SQL query.

Action 3: execute[SELECT contestant_name FROM contestants WHERE contestant_name != 'Jessie Alloway'] Observation 3: [('Edwina Burnam',), ('Tabatha Gehling',), ('Kelly Clauss',), ('Alana Bregman',), ('Jessie Eichman',), ('Allie Rogalski',), ('Nita Coster',), ('Kurt Walser',), ('Ericka Dieter',), ('Loraine NygrenTania Mattioli',)]

Thought 4: The SQL query worked, and the output shows all the contestant's names, but does not include 'Jessie Alloway'. I am finished with this query.

Action 4: submit

---

**ReAct_refined Example 2**

Question: What are the names of the contestants whose names are not 'Jessie Alloway'?

**Thought 1: I should first find out what tables are available in this MySQL database that can help me answer this question**

**Action 1: execute[SHOW TABLES]**

Observation 1: [('contestants')]

Thought 2: It looks like there is a table called 'contestants', but it doesn't have a name column. I need to find out what columns are available.

Action 2: execute[DESCRIBE contestants]

Observation 2: [('contestant_number', 'int', 'NO', 'PRI', None, 'auto_increment'), ('contestant_name', ...)]

Thought 3: The available columns for the 'contestant' table are 'contestant_number' and 'contestant_name'. I can now write a SQL query.

Action 3: execute[SELECT contestant_name FROM contestants WHERE contestant_name != 'Jessie Alloway']

...(same as original ReAct example)

---

Table 21: Example 2) Comparison of example trajectories from ReAct and refined version of ReAct for the SQL task.

