# OpenReview forum: "StateFlow: Enhancing LLM Task-Solving through State-Driven Workflows"
_colmweb.org/COLM/2024/Conference — COLM_

### Official Review · Reviewer_t4Kc · 2024-05-09

**Rating:** 7
**Confidence:** 5
**Ethics Flag:** 1

**Summary:**

This paper proposes a task-solving paradigm, StateFlow, which conceptualizes complex task-solving processes as a state machine. This approach separates the control of task decomposition from sub-task solving and achieves better performance with fewer interaction turns compared to baselines on the InterCode SQL and ALFWorld benchmarks.

**Questions To Authors:**

See Reasons To Reject.

**Reasons To Accept:**

1. LLM-based complex task-solving is a significant research area, and this author tackles the problem through a controllable paradigm design, which enhances effectiveness and has the potential to advance development in this field.
2. The proposed workflow achieves higher performance in two task scenarios while utilizing fewer interaction turns, demonstrating its utility.
3. This paper is well-written and includes all necessary details.
4. The experiments convincingly demonstrate performance improvements over various baselines, including ReAct and multi-agent architectures.

**Reasons To Reject:**

1. This workflow seems effective in complex task-solving, but for simpler tasks such as translations or reading comprehension, which are foundational abilities of language models, it may not offer significant improvements. This may also explain why the performance gain is greater on the ALFWorld benchmark than on the InterCode SQL benchmark.
2. This framework is not entirely independent of human efforts. To deploy StateFlow for a specific task, we must carefully define the States, which are central to adapting it for specific tasks, along with output functions and transitions. Designing states is challenging because it initially requires observing task-solving examples. With the basic workflow in mind, we must consider all possible scenarios that might arise during the process. Moreover, as the ablation experiments show, the omission of any essential state can lower performance. This task presents challenges not only for models but also for humans.
3. This framework primarily enhances process control and does not improve reasoning or subtask solving; therefore, common errors, such as tool parameter errors, still occur.

============================

After rebuttal:

Initially, I gave the paper a relatively high score due to its interesting method. If the authors can solve my questions, **I will raise the score to defend this work**. However, the authors just agree with my opinion but **do not make an effort to address my concerns**. Thus, I **decrease the score**.

After another rebuttal:

I have changed the score to 7 and did not raise the score due to the limitation of the paper in the method. However, I still think this work is interesting and deserves to be published.

---

> ### Author Rebuttal · Authors · 2024-05-29
>
> **> 1st Reject Point: This workflow seems effective in complex task-solving…**
>
> Thanks for your feedback. We admit that our method is limited to simple tasks, especially those that only require one single step. We kindly clarify our method is mainly focused on complex task-solving processes that require several steps or interactions with tools and environments. We will make this point more clear in our future version.
>
>
> **> 2nd Reject Point: This framework is not entirely independent of human efforts…**
>
> Thank you for your feedback. We agree that our method involves human effort and task observation.  We kindly argue that it is common across different tasks and environments, as seen in many related works [1, 2]. We are also aware of this concern and we provide clear guidance in Section 3.2.
> Additionally, the state's design is not a complex procedure. Specifically, it's not essential to account for every possible scenario; states can be designed to be broadly applicable to various situations. For instance, in the bash task, we created only Solve, Verify, and Error states despite the variety of questions and potential errors. While the ablation experiments show performance drops, it is important to note that these results still surpass the baselines with significantly less cost.
>
> [1] Yao, S., Zhao, J., Yu, D., Du, N., Shafran, I., Narasimhan, K., & Cao, Y. (2022). React: Synergizing reasoning and acting in language models. arXiv preprint arXiv:2210.03629.
>
> [2] Maciej Besta, Nils Blach, Ales Kubicek, Robert Gerstenberger, Lukas Gianinazzi, Joanna Gajda, Tomasz Lehmann, Michal Podstawski, Hubert Niewiadomski, Piotr Nyczyk, et al. Graph of thoughts: Solving elaborate problems with large language models. arXiv preprint arXiv:2308.09687, 2023.
>
> **> 3rd Reject Point: This framework primarily enhances process control…**
>
> Thanks for your feedback! We admit that our method focuses on process control by breaking down a complex workflow into state transitions and states. But it also provides the model with more specific and concise instructions for each action to enhance the model's reasoning abilities and reduce errors. We also want to note that our method is agnostic to reasoning works like ToTs and Reflexion to further improve reasoning and reduce tool parameter errors.

---

> ### Comment · Reviewer_t4Kc · 2024-06-04
> **Thank you for your response!**
>
> > We admit that our method is limited to simple tasks, especially those that only require one single step.
>
> So, it can not be extended to real-world difficult tasks?
>
> > We agree that our method involves human effort and task observation.
>
> How to improve it?
>
> > We admit that our method focuses on process control by breaking down a complex workflow into state transitions and states.
>
> How to solve it?
>
> Overall, you just agree with my opinion but do not solve my concern. Thus, I decrease the score.

---

> ### Author Response · Authors · 2024-06-04
> **Clarification to questions**
>
> Dear reviewer t4Kc,
>
> Thank you for your comments and questions. We believe there may have been some misunderstandings regarding our previous rebuttal responses. To clarify, we have provided additional justifications below. Could you please kindly re-evaluate our method based on the replies?
>
> > So, it can not be extended to real-world difficult tasks?
>
> We are sorry for the misunderstanding. In the original rebuttal, we were trying to say that our method **focuses on** complex tasks that require tools and interactions but **"is limited in"** simple tasks (not “limited to”). That means our methods may not have obvious advantages on simple one-hop questions. However, it is suitable to real-world difficult tasks.
>
> > How to improve it?
>
> As mentioned in our previous reply, we provide clear guidance in Section 3.2 to simplify the construction process and minimize human effort. Also, we mentioned that it is not necessary to consider all possible scenarios when designing states, as shown by our Bash experiment. Finally, as mentioned in Section 6 of our paper, to further reduce human effort, we can access running trajectories from a StateFlow model, and employ another LLM to iteratively add or remove states to automate the process.
>
> > How to solve it?
>
> As mentioned in our previous reply, our method actually improves the reasoning of LLMs and reduces tool parameter errors because the LLMs are instructed with specific instructions at each state, so that they can follow a single instruction at a time for more accurate and dedicated reply.
>
> Thanks,
>
> Authors

---

> > ### Comment · Reviewer_t4Kc · 2024-06-05
> > **Thanks for your reply**
> >
> > What is the difference between “be limited in” and “be limited to”? I think it’s the same thing:)
> > I would like us to stop dwelling on grammar and return to the questions in the paper. Does your response mean that your method can be applied to complex tasks but is the simple part of complex tasks? Or does it mean it works but you didn't experiment? The current reply is too confusing for me to understand. Could you explain it again?
> >
> >
> > As for the second question, I think it is acceptable. Thank you for your clarification this time.
> >
> >
> > As for the third question, your reply is very different from the first one, which makes me very confused. How do you plan to solve it? Now what you say seems to be a bunch of formulaic words, could you explain it with an example, which will help me better understand it?

---

> > > ### Author Response · Authors · 2024-06-05
> > > **Further Clarification on complex tasks**
> > >
> > > Thanks for your patience and allow us to clarify.
> > >
> > > ## Re 1st question:
> > > 1. Our method can solve complex tasks. That means solving the whole task.
> > > 2. We performed experiments on complex tasks. All the benchmarks we used are generally considered as complex benchmarks. They all require multiple steps and interactions with the environment.
> > >
> > >
> > > ## Re 3rd question:
> > > **Example of a complex task (from ALFWorld, used in our experiment)**: Household setting: You are in the middle of a room. Looking quickly around you, you see a cabinet 6, a cabinet 5, a cabinet 4, a cabinet 3, a cabinet 2, a cabinet 1, …. Your task is to: heat some apple and put it in garbagecan
> > >
> > > Note: to solve the task, we need to find the apple, heat it, and then put it garbagecan. During the process, the agent needs to interact with the environment, for example, go to a listed place and see if the apple is there. (Please refer to *Figure 10* in the Appendix for this example and solving process by our method)
> > >
> > >
> > > **Your original point**: This framework primarily enhances process control and does not improve reasoning or subtask solving; therefore, common errors, such as tool parameter errors, still occur.
> > >
> > > **Your question**: How do you plan to solve it?
> > >
> > >
> > > **Our reply:**
> > > 1. Our framework **already improves** reasoning and subtask solving, as well as reducing tool parameter errors.  Let’s use the ALFWorld example above for illustration.
> > >
> > >    - As noted, solving this task requires multi-steps, and the LLM needs to decide what to do next based on feedback from the environment.
> > >    - If you could take a look at Figure 4 in our paper, our method decomposes the process to different states. Within each state, a specific instruction is used to prompt the model. For example, In the “Pick” state, only instructions related to picking up the object are sent to LLM.
> > >    - Compared to using one instruction that asks the model to perform all actions, we only ask the model to perform one action in a state. So the LLM can follow the instructions better, and we expect less common errors.
> > >
> > > 2. To further improve reasoning, iterative refining methods can be easily added to our framework.
> > > 	- For example, *Reflexion* learns from past errors to improve reasoning, and we show that adding Reflexion can improve the performance with less incurred costs (See results in Figure 5).

---

> > > > ### Comment · Reviewer_t4Kc · 2024-06-05
> > > > **Thanks for your reply**
> > > >
> > > > Thank you for your response. Now, I think both are acceptable. Thank you for clarifying this time. I have changed the score to 7.

---

> > > > > ### Author Response · Authors · 2024-06-05
> > > > > **Thank you**
> > > > >
> > > > > Thank you for your effort in communicating with us. Please let us know if you have any other questions!

---

### Official Review · Reviewer_JAb2 · 2024-05-12

**Rating:** 7
**Confidence:** 3
**Ethics Flag:** 1

**Summary:**

The paper addresses the broad topic of LLM prompt efficiency, specifically in the context of task-oriented interactions with external tools. The paper's main contribution is to propose structuring interactions as a task-specific finite state machine, where the task is decomposed into subtasks (states) with a dedicated instruction and criteria for transitioning out of the state. This formalism is evaluated on two LLM model checkpoints (GPT-4-Turbo, GPT-3.5-Turbo) across three broad tasks: SQL and Bash tasks from the InterCode benchmark and ALFWorld, a text-adventure environment. The proposed method is compared against the InterCode benchmark methods, ReAct and Plan & Solve, reporting on success rate, error rate, and LLM usage/cost.

The paper is generally easy to read and well motivated. The key innovation of the paper is the development of a state machine-inspired decomposition approach to LLM prompting. While motivated by and demonstrated on tool interaction tasks, this concept could be adapted to others where the the broader request can be decomposed into subtasks that have individual success criteria. The primary demonstrated benefit is efficiency -- fewer inference calls with fewer tokens, thus less cost, to achieve competitive or better results.

The primary weakness of the the paper is in its structure. It spends a lot of time describing finite state machines from a formal basis and attempting to match elements of the StateFlow method to this. However, the paper does not motivate why this is necessary or what value there is in doing so, especially as it has to modify the original FSM formalism to account for elements specific to the LLM interaction setting. There is an interesting result here, but the desire to mathematically align with the FSM definition gets in the way. It should be enough to say that StateFlow is inspired by FSMs, provide a brief schematic, and then move on to the actual details of the experiments.

The paper acknowledges that StateFlow's success is entirely dependent on the implementer to define an FSM with associated instructions and transition criteria for the specific task. In its current form, this makes it more of a strategy for structuring solutions to problems than a general method. Finally, in the appendix, there are details to enable reproducibility, including providing code in GitHub.

======

Edit: After rebuttal, the decision has changed to accept based on promised changes

**Reasons To Accept:**

The idea of decomposing complex requests into a state-machine structure is worth promoting.

**Reasons To Reject:**

The paper should be revised first in order to be stronger. Too much of the paper is spent reviewing the formal definition of finite state machines and attempting to adapt that to the LLM setting. This should be removed or relegated to the appendix, and replaced with content already in the appendix around the experimental details and analysis conducted.

---

> ### Author Rebuttal · Authors · 2024-05-29
>
> **> 1st rejection point: The paper should be revised first in order to be stronger….**
>
> Thank you for your valuable feedback! We included detailed FSM definitions to provide a formal structure and connect our method to these concepts to aid understanding.  We appreciate your suggestion and will revise the paper to condense these sections. We'll also move more experimental details and analysis from the appendix to the main text to strengthen the paper in our future version.

---

> > ### Comment · Reviewer_JAb2 · 2024-06-05
> >
> > Thank you for responding. The promised changes should indeed improve the paper.

---

> > > ### Author Response · Authors · 2024-06-05
> > > **Thank you**
> > >
> > > Thank you for your review, please let us know if you have any other questions!

---

### Official Review · Reviewer_voLS · 2024-05-13

**Rating:** 6
**Confidence:** 3
**Ethics Flag:** 1

**Summary:**

StateFlow presents a structured approach by modeling task-solving as state machines, allowing for a systematic progression through tasks, which slightly enhances the efficiency and interpretability of LLMs. The method shows modest improvements in performance on benchmarks such as InterCode SQL and ALFWorld, suggesting potential benefits over existing models like ReAct.

**Reasons To Accept:**

The proposed method appears to perform well on tasks from both InterCode and ALFWorld. It also seems to reduce costs when compared with ReAct and ALFChat.

**Reasons To Reject:**

To validate the effectiveness of the method, it is recommended to conduct more experiments using other models. How does this method perform with open-source LLMs such as LLaMA3 and Mistral (e.g., 7B and 70B versions)?

---

> ### Author Rebuttal · Authors · 2024-05-29
>
> Thanks for your suggestions. We tested the ALFWorld dataset with Mistral-7b, Llama3-8b, Llama3-70b and show the results below.  Additionally, we tested the stronger 3-agent ALFChat for comparison.
>
> ## Mistral-7b
> | Method            | Accuracy | Cost  |
> |-------------------|----------|-------|
> | ALFChat (3 agents) | 1.5%     | 0.708 |
> | StateFlow         | 6%       | 2.03  |
>
>
> ## Llama3-8b
> | Method            | Accuracy | Cost  |
> |-------------------|----------|-------|
> | ALFChat (3 agents) | 50.7%    | 0.52  |
> | StateFlow         | 62.7%    | 0.38  |
>
> ## Llama3-70b
> | Method            | Accuracy | Cost  |
> |-------------------|----------|-------|
> | ALFChat (3 agents) | 83.6%    | 2.53  |
> | StateFlow         | 94%      | 1.83  |
>
> We notice that Mistral-7b struggles to complete the task using both methods. This suggests that the Mistral-7b model itself is not equipped to handle tasks within ALFWorld. Additionally, we observe that StateFlow outperforms 3-agent ALFChat while also requiring fewer computational resources. This highlights the effectiveness of our approach.

---

> > ### Comment · Reviewer_voLS · 2024-06-04
> >
> > Thanks for the author's response and effort. It is recommended to incorporate the results into the next version.

---

> > > ### Author Response · Authors · 2024-06-05
> > >
> > > Thank you for your review! We will incorporate this results in the next version.

---

### Official Review · Reviewer_af37 · 2024-05-15

**Rating:** 6
**Confidence:** 3
**Ethics Flag:** 1

**Summary:**

The paper introduces StateFlow, a framework for enhancing task-solving processes in LLMs by using finite state machines (FSMs) to model complex workflows. The primary idea is the combination of state-driven control and dynamic interaction with external tools, thus improving the efficiency and interpretability of LLMs in multi-step problem-solving tasks. By structuring the LLM's task-solving process into distinct states and transitions, it enables better control and guidance, improving performance metrics over baseline methods like ReAct.

**Questions To Authors:**

- It seems that the states required to be pre-defined. Considering that human cognitive processes are not strictly confined to finite states, can you provide more insights on how StateFlow performs in highly dynamic environments where the tasks or the nature of the interactions change over time? Specifically, how does StateFlow adapt when faced with tasks that have evolving goals or when unexpected changes occur in the environment?
- Any thoughts or ideas on some more complex tasks, e.g., SWE-bench or real-world Web interaction?

**Reasons To Accept:**

- The use of FSMs to structure task-solving processes helps in breaking down complex tasks into clearer, more manageable steps, which enhances the interpretability and "debuggability" of LLMs in complex scenarios.
- The paper presents empirical evidence where StateFlow outperforms existing methods like ReAct in specific benchmarks with lower cost, indicating its effectiveness in controlled scenarios.
- The potential ability to integrate StateFlow with other iterative and refining methods.

**Reasons To Reject:**

- For me, the main concern of this paper is that the core idea of applying state machines to manage LLM workflows seems to repackage a well-established concept without providing transformative novelty. In the context of LLMs, while the paper presents this as a novel methodology, lots of agent already consider the similar ideas, albeit without explicitly claiming the FSM label.
- It seems that the demonstrated benefits of StateFlow are primarily shown in "controlled" benchmarks, which might not fully convince of its broad applicability or superiority in diverse real-world scenarios.
- I can resonate with the authors' choice of baselines, as there are indeed many agent-based methods that are complex and difficult to implement. However, I feel that the choice of baselines might be slightly weak. I would have liked to see comparisons with other methods, many of which are referenced in the related work section. This would help to contextualize StateFlow's performance and innovative value more robustly within the current landscape.

---

> ### Author Rebuttal · Authors · 2024-05-29
>
> Thank you for your feedback and suggestions!
>
> **> 1st Point**
>
> We do admit that the concept is present in the domain of LLM agents. However, we kindly argue that StateFlow has a different research focus and both of us make contributions to the community.
> Specifically, StateFlow focuses on offering a formalized and systematic framework to construct LLM workflows, enhancing clarity and interpretability. While other works introduce specific workflows for agents. For example, LATs  unifies reasoning, acting, and planning with MCTS and ADAPT [4] explicitly plans and decomposes subtasks iteratively.
>
> **> 2nd Point**
>
> We acknowledge that benchmarks like ALFWorld could be perceived as “controlled” due to its structured tasks. But we would like to highlight that the Bash Task is derived from real-world pages, showcasing StateFlow’s effectiveness. We believe that the three diverse datasets we use adequately show the applicability of StateFlow.
>
> **> 3nd Point**
>
> Due to the limited rebuttal time, we are not able to provide the experiments for other baselines in such a short time, especially given that some methods lack available source code. We really appreciate your suggestion and will definitely include more baselines in the future version after we carefully implement them!
>
> **> 1st question**
>
> StateFlow has two mechanisms to handle highly dynamic environments. Specifically,
> (1) As mentioned in Section 3.2, states can be designed to be less specific (e.g., using a single state to handle all errors), making them more adaptable to dynamic environments.
>
> (2) StateFlow can also fit tasks with evolving goals by creating state transitions that detect changes in task objectives. When the change is detected, the system can transit to a handling state for the anomaly and alert developers to take appropriate action. The modular nature of StateFlow allows for easy modifications: new states can be added without altering the overall behavior, and pre-defined states can be reused by simply adding new connections.
>
> **> 2nd question**
>
> For complex tasks involving multiple steps and conditions, StateFlow provides a structured and simplified approach to state management. Specifically, (1) each condition is handled individually, making the system easier to debug and improve. (2) Running trajectories (such as those shown in Figure 3) can be easily retrieved for analysis, and failures can be examined (as detailed in Appendix B.2) to refine the StateFlow.

---

### Decision · Program_Chairs · 2024-07-10

**Decision:**

Accept

**Comment:**

This paper proposes a StateFlow framework to better control LLM to solve highly complex tasks. The method utilizes the concept of FSMs to model the task-solving process in LLMs, and proposes a few common components (observe, select, verify etc.) to help the model transit between different states, reaching a more correct final state for solving the task.

Pros:
- Reviewers agree that the proposed method is relatively novel and well-motivated: StateFlow helps LLMs break complex tasks into clearer, more manageable sub-steps. This could improve the controllability and interpretability of LLMs.
- The empirical gains are fairly significant over multiple tasks, with a reduced cost.

Cons:
- Most reviewers point out that the biggest weakness of this paper is that all the states need to be pre-defined and all the tasks experimented can be well-controlled. It is difficult to know whether such framework can be easily adapted to real-world complex tasks.

Minor:
- Some reviewers mentioned that the structure of this paper can be made more clear, like the description of FSMs can be condensed and the authors should provide more details and analysis for the method and experiments.